# Conformal Generative Modeling with Improved Sample Efficiency through Sequential Greedy Filtering

**Klaus-Rudolf Kladny[1,2], Bernhard Schölkopf[1,2,3], Michael Muehlebach[1]**
[1]Max Planck Institute for Intelligent Systems, Tübingen
[2]Tübingen AI Center
[3]ELLIS Institute Tübingen
{kkladny, bs, michaelm}@tue.mpg.de

## Abstract

Generative models lack rigorous statistical guarantees for their outputs and are therefore unreliable in safety-critical applications. In this work, we propose *Sequential Conformal Prediction for Generative Models* (*SCOPE-Gen*), a sequential conformal prediction method producing prediction sets that satisfy a rigorous statistical guarantee called conformal admissibility control. This guarantee states that with high probability, the prediction sets contain at least one admissible (or valid) example. To this end, our method first samples an initial set of i.i.d. examples from a black box generative model. Then, this set is iteratively pruned via so-called greedy filters. As a consequence of the iterative generation procedure, admissibility of the final prediction set factorizes as a Markov chain. This factorization is crucial, because it allows to control each factor separately, using conformal prediction. In comparison to prior work, our method demonstrates a large reduction in the number of admissibility evaluations during calibration. This reduction is important in safety-critical applications, where these evaluations must be conducted manually by domain experts and are therefore costly and time consuming. We highlight the advantages of our method in terms of admissibility evaluations and cardinality of the prediction sets through experiments in natural language generation and molecular graph extension tasks.

## 1 Introduction

Generative models have found extensive applications across various domains, including the generation of images (Rombach et al., 2022), molecules (Vignac et al., 2023), and natural language (Achiam et al., 2023). While these models are becoming increasingly performant, caution is required when decisions are based on generated outputs. This is particularly relevant for scenarios where errors have severe consequences. A prime example is the phenomenon of "confabulations" (or "hallucinations") in large language models—responses that lack factual support, potentially leading to substantial harm if trusted (Bai et al., 2024; Weidinger et al., 2021). Therefore, developing effective measures to ensure strong statistical guarantees on the predictions of these models is imperative.

The field devoted to obtaining rigorous statistical guarantees for the predictions of machine learning models is called *risk control* (Angelopoulos et al., 2022; 2021; Bates et al., 2021). A prominent line of work within this field is called *conformal prediction* (Vovk et al., 2005; Shafer & Vovk, 2008). Conformal prediction techniques typically use a heuristic uncertainty estimate from a black box machine learning model to construct a non-conformity measure (e.g., Kato et al. 2023) that allows for generating prediction sets by selecting examples in the prediction space $\mathcal{Y}$ based on their non-conformity values. A typical guarantee is that with high probability, the true label $y \in \mathcal{Y}$ (as sampled from the generative process) for a test example $x$ is contained in this prediction set. Conformal prediction is appealing due to its simplicity and strong non-asymptotic, distribution-free guarantees (Angelopoulos & Bates, 2021). However, conformal prediction methods are typically designed for classification or low-dimensional regression tasks and they suffer from the curse of

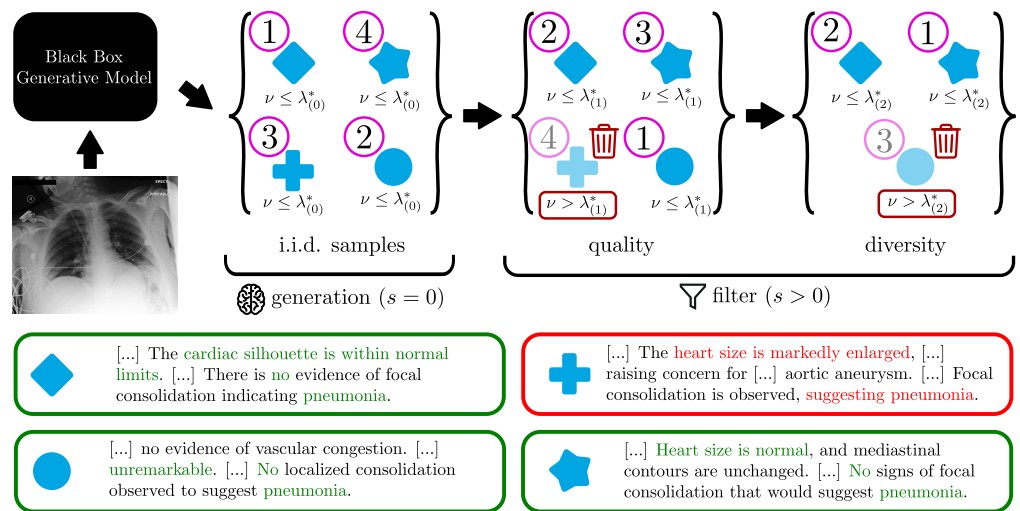

Figure 1: **SCOPE-Gen for Radiology Report Generation.** The sequential prediction procedure can be separated into two stages: In the generation stage ($s = 0$), i.i.d. text reports (blue tokens) are drawn from the generative model. In the filter stage ($s \in \{1, 2\}$), the prediction set from the previous step $s - 1$ is refined to remove examples with low quality (removing the false response in red) and examples with high similarity to already sampled texts (the answers in green are similar). Each step is performed via iterative (sub-)sampling in a given order (indicated by the purple circles), until the value of a non-conformity variable $\nu$ exceeds a calibrated threshold $\lambda^*_{(s)}$.

dimensionality when the prediction space $\mathcal{Y}$ is combinatorially large or infinite, as it is the case for drug design or natural language generation.

Quach et al. (2024) have recently introduced a method called *conformal language modeling* (CLM) for addressing this problem. [1] Rather than evaluating a non-conformity measure on each point in the prediction space (which is intractable for textual data), CLM calibrates a *stopping rule* parameter $\lambda_{(0)}$ that decides how many samples $y$ are required from a generative model $G$, conditionally on features $x$ (for example, an image or a text query). These samples then make up the prediction set, which fulfills a statistical guarantee with respect to containing at least one admissible (correct/non-hallucinated) example. The method further introduces rejection sampling to ensure that the prediction set satisfies desired properties. The sampling is controlled by two calibration parameters $\lambda_{(1)}$ and $\lambda_{(2)}$ that ensure

1. Quality: If a new sample has too low quality, it is rejected. The quality threshold is given by $\lambda_{(1)}$.

2. Diversity: If a new sample from the generative model is too similar to an already sampled one, it is rejected. The similarity threshold is given by $\lambda_{(2)}$.

Conformal prediction cannot be applied with multi-dimensional calibration parameters, as noted by Angelopoulos et al. (2022). To address this limitation, CLM calibrates all parameters $\boldsymbol{\lambda} := (\lambda_{(0)}, \lambda_{(1)}, \lambda_{(2)})$ using Pareto testing (Laufer-Goldshtein et al., 2023), a technique derived from the learn-then-test framework with fixed sequence testing for family-wise error rate control (Angelopoulos et al., 2021). In broad strokes, Pareto testing divides the calibration data into two sets: The first set is used to estimate the Pareto frontier of different configurations for $\boldsymbol{\lambda}$ over mean prediction set size (to be minimized) and admissibility (to be controlled). The estimated Pareto frontier is used to construct a fixed sequence whose elements are tested in order, on the second data set. For testing, the null hypothesis corresponds to not controlling the admissibility at a given level. The returned solution for $\boldsymbol{\lambda}$ is the last configuration in the sequence for which this null hypothesis is rejected.

While CLM offers a notion of admissibility control in generative models, it has a practical limitation. Specifically, during calibration, CLM necessitates sampling and evaluating admissibility for a fixed number of times per instance $x$ in the calibration set. In safety-critical applications, such as medicine, the factual integrity of admissibility checks based on n-gram (Lin, 2004; Papineni et al., 2002) or

---

[1]Notably, despite its name, CLM is not based on conformal prediction in a strict sense.

machine learning methods (Smit et al., 2020) remains a concern (Zhang et al., 2024b), necessitating manual assessments by human domain experts for accurate calibration. Reducing the number of these costly admissibility checks is thus paramount (Settles, 2009; Mosqueira-Rey et al., 2023).

Recognizing the need to enhance efficiency, we propose a method called *Sequential Conformal Prediction for Generative Models* (SCOPE-Gen), visualized in Fig. 1. This method differs in two major ways from CLM: **(1)** Prediction set generation is performed sequentially and can be split up into an initial generation stage and a filter stage. The generation stage proposes i.i.d. candidates from the generative model (blue tokens in Fig. 1), the filter stage applies so-called *greedy filters* in order to sub-sample the prediction set from the previous step. **(2)** We apply conformal prediction instead of learn-then-test. This can be done due to the fact that the admissibility of our final prediction set factorizes as a Markov chain. In this chain, each factor can be controlled separately. Thus, the desired level of admissibility for the final prediction set can be controlled by separately calibrating the individual prediction step parameters according to the Markov chain factorization.

Importantly, our prediction procedure has the significant advantage of *reducing the number of admissibility checks* involved in *oracle-in-the-loop*[2] scenarios (we explain this in detail in § 4.2).

In summary, we highlight the following contributions:

- We introduce *Sequential Conformal Admissibility Control for Generative Models* (*SCOPE-Gen*), a sequential conformal prediction approach that satisfies admissibility control (§ 2) for black box generative models (§ 4).
- SCOPE-Gen requires significantly fewer queries of an admission function, in comparison to prior work. In addition, its computational complexity is lower ($\mathcal{O}(K)$ in comparison to $\mathcal{O}(g^K)$, where $K$ is the number of calibration parameters (typically $K = 3$) and $g$ is the size of a parameter grid).
- We experimentally demonstrate the advantages of SCOPE-Gen in comparison to various baselines and prior work on molecular scaffold extension and natural language generation tasks (§ 5).

**Overview.** Section § 2 introduces (conformal) admissibility control, which is the main objective of the present work. Thereafter, we present conformal prediction in Section § 3. These sections set the stage for presenting our method in Section § 4. Upon establishing the main principles, we demonstrate the sub-sampling and non-conformity update functions used for filtering (§ 4.1), the algorithmic details of the calibration (§ 4.2). Thereafter, we experimentally demonstrate the advantages of our approach in comparison to prior work in Section § 5, followed by a section dedicated to related work (§ 6). Finally, we conclude in Section § 7. A discussion section is provided in App. I.

## 2 OBJECTIVE

We consider generating a prediction set $\mathcal{C}_{\boldsymbol{\lambda}}(X) = \{Y_i\}$, parameterized by a calibration parameter $\boldsymbol{\lambda} \in \mathbb{R}^K$ computed from a labeled i.i.d. calibration set $\{(X_i, Y_i^t)\}_{i=1}^n$ with feature variable $X_i$ and ground truth prediction $Y_i^t$. The main objective of the present work is for $\mathcal{C}_{\boldsymbol{\lambda}}$ to be admissible for a new example, with high probability. Formally, we say that $\boldsymbol{\lambda}$ satisfies *(conformal) admissibility control* at a given level $\alpha \in (0,1)$ if

$$\mathbb{P}(\exists Y \in \mathcal{C}_{\boldsymbol{\lambda}}(X_{n+1}) : a(Y, Y_{n+1}^t) = 1) \geq 1 - \alpha, \tag{1}$$

where $(X_{n+1}, Y_{n+1}^t)$ is an additional independent sample from the same distribution as the calibration set and $a : \mathcal{Y}^2 \mapsto \{0, 1\}$ (no additional assumptions on $a$ are imposed) measures admissibility between generated sample $Y \in \mathcal{Y}$ and $Y_{n+1}^t \in \mathcal{Y}$ (Fisch et al., 2021)[3]. For instance, $a$ could measure whether a textual response from a language model $Y$ for the input $X$ is *similar* to a ground truth textual response $Y_{n+1}^t$. For completeness, we demonstrate the relationship between admissibility control (1) and the more general notion called *conformal risk control* (Angelopoulos et al., 2022) in App. A. For notational convenience, we introduce the *set-level admissibility*

$$A(\mathcal{C}_{\boldsymbol{\lambda}}(x), y^t) := \mathbb{1}\{\exists y \in \mathcal{C}_{\boldsymbol{\lambda}}(x) : a(y, y^t) = 1\}. \tag{2}$$

We will generally drop dependence of $A$ on $x$ and $y^t$ and simply write $A(\mathcal{C}_{\boldsymbol{\lambda}})$ instead of $A(\mathcal{C}_{\boldsymbol{\lambda}}(x), y^t)$. We also introduce the *(total) admissibility*

$$\mathcal{A}(\boldsymbol{\lambda}) := \mathbb{P}(A(\mathcal{C}_{\boldsymbol{\lambda}}(X_{n+1}), Y_{n+1}^t)), \tag{3}$$

---

[2]This means that an oracle can be queried on demand by the algorithm.

[3] We note that $Y_{n+1}^t$ is not observed at test time.

which is the quantity we desire to bound from below by $1 - \alpha$.

## 3 PRELIMINARY : CONFORMAL PREDICTION

Conformal prediction (specifically, *split conformal prediction* (Papadopoulos et al., 2002; Lei et al., 2015)) provides a way of generating a prediction set $\mathcal{C}_\lambda(x) = \{y_i\}$, parameterized by a one-dimensional calibration parameter $\lambda \in \mathbb{R}$. The typical setup in conformal prediction is to control *(marginal) coverage*, which is defined as

$$\mathbb{P}(Y_{n+1}^t \in \mathcal{C}_\lambda(X_{n+1})) \geq 1 - \alpha. \tag{4}$$

Conformal prediction techniques assume a (possibly random) non-conformity measure $\nu(x, y)$, which is typically chosen as a heuristic estimate for how incompatible $x$ and $y$ are. For instance, if $\mathcal{Y}$ is a finite set of labels and $X$ corresponds to the feature variable (e.g., an image), a simple choice for $\nu$ is to consider the softmax scores of a classifier and use $\nu(x, y) = 1 - \texttt{softmax}(x, y)$. Given a choice of non-conformity $\nu$ and a calibration parameter $\lambda$, the prediction set is constructed as

$$\mathcal{C}_\lambda(x) := \{y \in \mathcal{Y} : \nu(x, y) \leq \lambda\}. \tag{5}$$

It can be shown that (4) is satisfied (independent of the choice of $\nu$) when calibrating $\lambda$ as follows:

$$\lambda^* \left(\{\nu(x_i, y_i)\}_{i=1}^n ; \alpha\right) = \widehat{\text{quantile}} \left(\{\nu(x_i, y_i)\}_{i=1}^n ; \frac{\lceil (1 - \alpha)(n + 1) \rceil}{n}\right), \tag{6}$$

where $\widehat{\text{quantile}}$ denotes the empirical quantile of $\nu$, evaluated at the calibration points $\{\nu(x_i, y_i)\}_{i=1}^n$ [4]. We provide a rigorous definition of the empirical quantile (6) in App. B .

We note that (4) can be seen as a special case of admissibility control (1), where the function $a(y, y^t)$ (2) returns 1 if and only if $y$ and $y^t$ are identical. In the current study, we extend this concept to accommodate the typically vast output space $\mathcal{Y}$ — for instance, encompassing all possible sentences in natural language. This extension allows the function $a$ to accommodate cases where $y$ and $y^t$ are not identical but are semantically equivalent or similar. Without such a generalization, prediction sets would need to be excessively large to ensure admissibility control at the desired level (1), since they would have to account for all possible variations in structuring a sentence.

## 4 SEQUENTIAL CONFORMAL PREDICTION FOR GENERATIVE MODELS

SCOPE-Gen relies on i.i.d. calibration data $\{(x_i, y_i^t)\}_{i=1}^n$ from the true generating process. In order to control $\mathcal{A}(\boldsymbol{\lambda})$ (3) at level $\alpha$, we utilize conformal prediction § 3. The main challenge of our setting is that the output space $\mathcal{Y}$ is combinatorially large or even infinite. Hence, the prediction set determined as in (5) (with $\boldsymbol{\lambda} = \lambda \in \mathbb{R}$) is intractable. Thus, we construct a prediction set directly from i.i.d. samples from a generative model $G$.

**Sequential Prediction (Alg. 1).**   The underlying idea of SCOPE-Gen is to construct the prediction set $\mathcal{C}_{\boldsymbol{\lambda}}$ in multiple sequential steps $s = 0, 1, 2$, as shown in Alg. 1 and Fig. 1. On a higher level, this process can be divided into two stages:

1. Generation Stage ($s = 0$): An initial prediction set $\mathcal{C}_{(0)}$ is generated by iteratively adding new i.i.d. samples from a generative model $G$, conditioned on $x$ (line 5, line 13). A new sample $y$ is added until a non-conformity value $\nu$, which is updated at each iteration of the while loop (line 10), exceeds the value of the calibration parameter $\lambda_{(0)}$ (line 12). We assume that $\nu$ is increasing at each iteration.

2. Filter Stage ($s > 0$): A generated prediction set $\mathcal{C}_{(s-1)}$ is filtered (or pruned) to $\mathcal{C}_{(s)}$, as to further reduce prediction set size (i.e., $|\mathcal{C}_{(s)}| \leq |\mathcal{C}_{(s-1)}|$). In this stage, filters $s = 1, 2$ are be applied sequentially. Each filter is parameterized by an individual calibration parameter $\lambda_{(s)}$ and is computed by means of a sub-sampling operation $\texttt{sub\_sample}(\mathcal{C}_{(s)}, \mathcal{C}_{(s-1)})$. This sub-sampling is performed (line 9) until the non-conformity value $\nu$ exceeds $\lambda_{(s)}$ (line 12; analogous to the generation stage) or there are no more samples left to add (line 8).

---

[4] Intuitively, $\lambda^*$ is the quantile at level $\approx 1 - \alpha$, with a small correction for theoretical reasons (e.g., Angelopoulos & Bates (2021)).

---

**Algorithm 1:** `predict`

**Input:** Condition $x$; generative model $G$; sub-sampling functions $\{\texttt{sub\_sample}_{(s)}\}_{s=1}^2$;
update functions $\{\texttt{update}_{(s)}\}_{s=1}^2$, calibration parameters $\{\lambda_{(s)}\}_{s=0}^2$;

1   **for** $s = 0, 1, 2$ **do**
2     $\mathcal{C}_{(s)}^0 \leftarrow \varnothing; \nu \leftarrow 0; j \leftarrow 0$               *// initialization*
3     **while** *true* **do**
4       **if** $s = 0$ **then**
5         $y \sim G \,|\, x$                *// generation stage*
6       **else**
7         **if** $\mathcal{C}_{(s)}^j = \mathcal{C}_{(s-1)}$ **then**
8           **break**               *// no samples left*
9         $y \sim \texttt{sub\_sample}_{(s)}(\mathcal{C}_{(s)}^j, \mathcal{C}_{(s-1)})$       *// filter stage*
10      $\nu \leftarrow \texttt{update}_{(s)}(\nu, y, j)$
11      **if** $\nu > \lambda_{(s)}$ **then**
12        **break**               *// threshold reached*
13      $\mathcal{C}_{(s)}^{j+1} \leftarrow \mathcal{C}_{(s)}^j \cup \{y\}$
14      $j \leftarrow j + 1$
15 **return**   $\mathcal{C}_{(2)}$

---

This sequential prediction provides a remedy for the intractability of the conformal prediction set as computed in (5). However, it poses the challenge that the total admissibility $\mathcal{A}(\boldsymbol{\lambda})$ is parameterized by a multi-dimensional calibration parameter $\boldsymbol{\lambda}$, which cannot be directly calibrated via conformal prediction (6). This is due to the fact that conformal prediction crucially depends on the assumption that the loss is controlled by a one-dimensional parameter (Angelopoulos et al., 2022).

**Markov Chain Factorization.** We address this issue by noting that in our sequential prediction procedure, the total admissibility $\mathcal{A}(\boldsymbol{\lambda})$ factorizes as a Markov chain

$$\mathcal{A}(\boldsymbol{\lambda}) = \underbrace{\mathbb{P}(A(\mathcal{C}_{\lambda_{(0)}}) = 1)}_{=: \,\mathcal{A}(\lambda_{(0)})} \underbrace{\mathbb{P}(A(\mathcal{C}_{\lambda_{(1)}}) = 1 \,|\, A(\mathcal{C}_{\lambda_{(0)}}) = 1)}_{=: \,\mathcal{A}(\lambda_{(1)})} \underbrace{\mathbb{P}(A(\mathcal{C}_{\lambda_{(2)}}) = 1 \,|\, A(\mathcal{C}_{\lambda_{(1)}}) = 1)}_{=: \,\mathcal{A}(\lambda_{(2)})}. \quad (7)$$

This means that the probability that the final prediction set $\mathcal{C}_{(2)}$ contains an admissible example is determined by the probability that the first step generates at least one admissible example and the probabilities that none of the consecutive steps filter out all admissible examples. Consequently, we can control the total admissibility $\mathcal{A}(\boldsymbol{\lambda})$ at level $\alpha$ by controlling each individual admissibility $\mathcal{A}(\lambda_{(s)})$ for univariate $\lambda_{(s)} \in \mathbb{R}$ and for $s \in \{0, 1, 2\}$, separately, where the individual levels $\alpha_{(s)}$ are chosen such that $\alpha = 1 - \prod_{s=0}^2 (1 - \alpha_{(s)})$. We discuss the choice of $\alpha_{(s)}$ in App. D.1.

**Employing Conformal Prediction.** We now proceed to show how we use conformal prediction to control the individual admissibilites $\mathcal{A}(\lambda_{(s)})$. First, we note that the prediction sets in SCOPE-Gen are created at each step $s$ such that they grow incrementally, meaning that $\mathcal{C}_{(s)}^j \subset \mathcal{C}_{(s)}^{j+1}$, where $\mathcal{C}_{(s)}^j$ is the prediction set $\mathcal{C}_{(s)}$ after $j$ iterations of sampling in a while loop of Alg. 1. As a result, we conclude that admissibility can be reformulated for $s = 0$ as

$$\mathbb{P}(A(\mathcal{C}_{(0)}^j) = 1) = \mathbb{P}(\text{``}l \leq j \text{ samples are required to satisfy admissibility''}), \quad (8)$$

and analagously for $s > 0$. Thus, just as in vanilla conformal prediction (5), we generate the prediction set over integers that controls the right-hand side of (8)

$$\tilde{\mathcal{C}}_{(s)}(x) = \{l \in \mathbb{N} : \nu(x, l) \leq \lambda^*\} = \{1, 2, ..., j\}, \quad (9)$$

where $\lambda^*$ is the calibrated parameter (6) and we recall that $\nu$ is increasing in $l$, for fixed $x$.[5] We see that sampling the prediction set $\mathcal{C}_{(s)}$ by sampling $\arg\max_l \tilde{\mathcal{C}}_{(s)}$ many examples controls the individual admissibility $\mathcal{A}(\lambda_{(s)})$ at level $\alpha_{(s)}$, which is equivalent to adding new samples to $\mathcal{C}_{(s)}$ until the non-conformity measure $\nu$ exceeds $\lambda_{(s)}$, as done in Alg. 1.

---

[5]We note that computing $\tilde{\mathcal{C}}_{(s)}(x)$ in (9) would generally be intractable for non-increasing $\nu$.

**Non-Conformity Measures.**  A crucial aspect for the performance of conformal prediction methods lies in the choice of the non-conformity measure, which is parameterized by the `update` function in Alg. 1. Here, we only discuss the choice of this function at the generation level (i.e., $\text{update}_{(0)}$) and refer to § 4.1 for the non-conformity measures used during filtering. For generation, we take inspiration from Quach et al. (2024) and consider the following update functions:

$$
\text{update}(\nu, y, j) = \begin{cases} j, & \text{(count)} \\ \nu \; + \; q(y) \; + \; \gamma j, & \text{(sum)} \\ \max\{\nu, \; q(y)\} \; + \; \gamma j, & \text{(max)} \end{cases} \tag{10}
$$

where $\gamma > 0$ and $q$ with $q(y) > 0, \forall y$ denotes a quality estimate of point $y$, such as an estimate of $p(y|x)$ by the generative model. We note that the *sum* and *max* updates are slightly different from Quach et al. (2024) due to the $+\gamma j$ term. The reason is that our prediction sets are not naturally limited in terms of size. Without this size regularization, prediction sets could become large.

## 4.1 GREEDY FILTERS

A core ingredient of SCOPE-Gen lies in the greedy filters. The term *greedy* reflects the incremental growth of the set of filtered points, $\mathcal{C}_{(s)}$, over the iterations $j$ in Alg. 1. We consider two types of filters: 1) quality filter and 2) diversity filter. Both of them are defined via a sub-sampling function `sub_sample` and a non-conformity update function `update`.

**Diversity Filter.**  We take inspiration from farthest point (sub-)sampling, which is common in computer vision applications (e.g., Moenning & Dodgson (2003); Qi et al. (2017)):

$$
\text{sub\_sample}(\mathcal{C}_{(s)}, \mathcal{C}_{(s-1)}) = \begin{cases} \sim \text{Uniform}(\mathcal{C}_{(s-1)}), & \text{if } \mathcal{C}_{(s)} = \varnothing \\ \arg\max_{y' \in \mathcal{C}_{(s-1)} \backslash \mathcal{C}_{(s)}} \min_{y'' \in \mathcal{C}_{(s)}} d(y', y''), & \text{else.} \end{cases}
$$

Intuitively, this method chooses the point in $\mathcal{C}_{(s-1)} \backslash \mathcal{C}_{(s)}$ with the maximum distance (as measured by some distance function $d$) from the already sampled subset $\mathcal{C}_{(s)}$, if $\mathcal{C}_{(s)}$ is non-empty. For empty $\mathcal{C}_{(s)}$, it chooses a point of $\mathcal{C}_{(s-1)}$, uniformly at random. For this filter, the update rule is specified as

$$
\text{update}(\nu, y) = \begin{cases} -d_{\max}, & \text{if } \nu = 0 \\ - \min_{y' \in \mathcal{C}_{(s)} \backslash \{y\}} d(y, y'), & \text{else,} \end{cases}
$$

where $d_{\max}$ is an upper bound on the distance function (see App. D.2 for details).

**Quality Filter.**  Using a quality function $q$, we choose the point $\mathcal{C}_{(s-1)} \backslash \mathcal{C}_{(s)}$ with maximum quality:

$$
\text{sub\_sample}(\mathcal{C}_{(s)}, \mathcal{C}_{(s-1)}) = \underset{y' \in \mathcal{C}_{(s-1)} \backslash \mathcal{C}_{(s)}}{\text{argmax}} \; q(y').
$$

For the update function of the non-conformity, we introduce the natural choice

$$
\text{update}(y) = -q(y).
$$

By default, if both filters are applied, we first apply the diversity filter.

## 4.2 CALIBRATION

For calibration, we split the i.i.d. calibration data $\{(x_i, y_i^t)\}_{i=1}^n$ into three non-overlapping subsets:

$$
\left\{(x_i, y_i^t)\right\}_{i=1}^{n_{(0)}}, \; \left\{(x_i, y_i^t)\right\}_{i=n_{(0)}}^{n_{(1)}}, \; \left\{(x_i, y_i^t)\right\}_{i=n_{(1)}}^{n_{(2)}},
$$

where $n_{(2)} = n$. This step is paramount, since otherwise, the admissibility controls in (7) are not independent. The calibration happens sequentially ($s = 0, 1, 2$), similarly to the prediction step, where calibration for $\lambda_{(s)}$ is performed using $\{(x_i, y_i^t)\}_{i=n_{(s-1)}}^{n_{(s)}}$. We now proceed to explain the algorithmic implementation of the generation and filter calibration, displayed in Alg. 2 and Alg. 3.

$y_i^{(1)}$  $y_i^{(2)}$  $\mathbf{y_i^{(3)}}$  $\left( y_i^{(4)} \quad \cdots \quad y_i^{(\text{max})} / y_i^{(|\mathcal{C}_{(s-1)}|)} \right)$

❌ ❌ ✅ ❌ ✅

❌ $a(y_i^{(j)}, y_i^t) = 0$   ✅ $a(y_i^{(j)}, y_i^t) = 1$

**(a)** generation:
$$y_i^{(j)} \overset{\text{i.i.d.}}{\sim} G|x_i$$

**(b)** filter:
$$y_i^{(j)} \sim \texttt{sub\_sample}(\mathcal{C}_{(s-1)}(x_i))$$

Figure 2: **SCOPE-Gen Savings on Admissibility Checks.** Samples in brackets denote samples that do not need to be assessed for admissibility (the ones sampled after the first admissible one). For the generation step **(a)**, the $y_i^{(j)}$ are i.i.d. samples from the generative model $G$. For filter steps **(b)**, the $y_i^{(j)}$ are examples from the previous prediction set $\mathcal{C}_{(s-1)}(x_i)$, ordered according to $\texttt{sub\_sample}$.

**Generation Calibration (Alg. 2).** As outlined (§ 4), for each $x_i$, we generate i.i.d. samples $y_i^{(j)}$ from $G \,|\, x_i$, until the first $y_i^{(j)}$ is admissible (line 3). The resulting non-conformity values $\nu_i$ are then used to calibrate $\lambda_{(0)}$. In practice, we typically need to limit the maximum amount of samples drawn by $\max \in \mathbb{N}$ (line 4) to tackle "hard" cases, i.e., examples $(x_i, y_i^t)$ where the generative model is unable to generate an admissible prediction. For these examples, we cannot generate finite non-conformity values and set these values to $\infty$ (line 13). Provided that such examples occur infrequently, the conformal quantile $\lambda_{(0)}^*$ (6) is not affected. Otherwise, SCOPE-Gen rejects the calibration[6], as common for methods in the field of risk control (e.g., Angelopoulos et al. (2021)).

**Filter Calibration (Alg. 3).** The calibration of $\lambda_{(s)}$, $s > 0$ is analogous to the calibration of the generation step (Alg. 2). However, we see from the factorization in (7) that filter steps must be calibrated on the condition that the prediction set of the previous step is admissible ($A(\mathcal{C}_{\lambda_{(s-1)}}) = 1$). Thus, we require sampling from the distribution with density

$$P(\mathcal{C}_{(s-1)} \,|\, A(\mathcal{C}_{(s-1)}) = 1) \propto \mathbb{1}\{A(\mathcal{C}_{(s-1)}) = 1\}P(\mathcal{C}_{(s-1)}), \tag{11}$$

where $P(\mathcal{C}_{(s-1)})$ denotes the density of the prediction set $\mathcal{C}_{(s-1)}$ as calibrated in the earlier step $s - 1$. In practice, we sample from (11) through the acceptance-rejection method. To this end, we first sample a prediction set for $s - 1$ (line 3). Instead of assessing its admissibility right away (leading to $|\mathcal{C}_{(s-1)}|$ admissibility checks), we can be more efficient by integrating these checks into the evaluation of the non-conformity value $\nu_i$. Specifically, we gradually add samples to $\mathcal{C}_{(s)}$ using $\texttt{sub\_sample}_{(s)}$ (line 8). Once an admissible example is encountered, no more checks need to be performed because we know that $\mathcal{C}_{(s-1)}$ is admissible. If $\mathcal{C}_{(s-1)}$ is inadmissible, the sub-sampling is repeated until there are no more samples to add, i.e., $|\mathcal{C}_{(s)}| = |\mathcal{C}_{(s-1)}|$ (line 6). In this case, no non-conformity value is generated (line 12).

For all calibration steps, we thus only require to perform repeated sampling until the first admissible example is encountered, as visualized in Fig. 2.

## 5 EXPERIMENTS

We compare our method against prior work and derived baselines to empirically demonstrate the advantages of SCOPE-Gen on real-world natural language tasks and a molecular generation task. As the motivation of the present work lies in sample efficiency, we generally focus on small calibration sets. The experiments in the main text consider the *sum* non-conformity measure with $\gamma = 0.5$ (10). We show qualitative evaluations in App. G, further quantitative evaluations and ablation studies in App. F.2 and an analysis of empirical admissibility (3) in App. F.1.

We compare our method against CLM (Quach et al., 2024) and a baseline derived from it. CLM controls a different notion of admissibility than we do. Specifically, CLM controls

$$\mathbb{P}\left[\mathbb{P}\left(A(\mathcal{C}_\lambda) = 1 \,|\, D_{\text{cal}}\right) \geq 1 - \beta_{(1)}\right] \geq 1 - \beta_{(2)}, \tag{12}$$

where $D_{\text{cal}}$ is shorthand for the calibration data $\{(x_i, y_i^t)\}_{i=1}^n$. We note that (12) controls admissibility (1) at level $\alpha$ if $\alpha = \beta_{(1)} + \beta_{(2)} - \beta_{(1)}\beta_{(2)}$ (we show this in App. C). In order to tighten the gap,

---

[6]which amounts to returning the entire output space $\mathcal{Y}$.

**Common Input:** Data $\{(x_i, y_i^t)\}_{i=n_{(s-1)}}^{n_{(s)}}$; admission function $A$; update function `update`

| **Algorithm 2:** `calibrate_generation` | **Algorithm 3:** `calibrate_filter` |
|---|---|
| **Input:** Generative model $G$ | **Input:** Sub-sampling func. `sub_sample`$_{(s)}$ |

**Algorithm 2: calibrate_generation**

**Input:** Generative model $G$

1 **for** $i = 1, 2, ..., n_{(0)}$ **do**
2 $\quad \mathcal{C}_{(0)}^0 \leftarrow \varnothing; \nu \leftarrow 0; j \leftarrow 0$
3 $\quad$ **while** $A(\mathcal{C}_{(0)}^j) = 0$ **do**
4 $\quad\quad$ **if** $|\mathcal{C}_{(0)}^j| = max$ **then**
5 $\quad\quad\quad$ **break**
6 $\quad\quad y^{(j)} \sim G \,|\, x_i$
7 $\quad\quad \nu \leftarrow \text{update}(\nu, y^{(j)}, j)$
8 $\quad\quad \mathcal{C}_{(0)}^{j+1} \leftarrow \mathcal{C}_{(0)}^j \cup \{y^{(j)}\}$
9 $\quad\quad j \leftarrow j + 1$
10 $\quad$ **if** $|\mathcal{C}_{(0)}^j| < max$ **then**
11 $\quad\quad \nu_i \leftarrow \nu$
12 $\quad$ **else**
13 $\quad\quad \nu_i \leftarrow \infty$
14 $\lambda_{(0)} \leftarrow \lambda^* \left(\{\nu_i\}_{i=1}^{n_{(0)}} ; \alpha_{(0)}\right)$
15 **return** $\lambda_{(0)}$

**Algorithm 3: calibrate_filter**

**Input:** Sub-sampling func. `sub_sample`$_{(s)}$

1 $m \leftarrow 0$
2 **for** $i = n_{(s-1)}, n_{(s-1)} + 1, ..., n_{(s)}$ **do**
3 $\quad \mathcal{C}_{(s-1)} \sim P(\mathcal{C}_{(s-1)} \,|\, x_i)$
4 $\quad \mathcal{C}_{(s)}^0 \leftarrow \varnothing; \nu \leftarrow 0; j \leftarrow 0$
5 $\quad$ **while** $A(\mathcal{C}_{(s)}^j) = 0$ **do**
6 $\quad\quad$ **if** $|\mathcal{C}_{(s)}^j| = |\mathcal{C}_{(s-1)}|$ **then**
7 $\quad\quad\quad$ **break**
8 $\quad\quad y^{(j)} \sim \text{sub\_sample}_{(s)}(\mathcal{C}_{(s)}^j, \mathcal{C}_{(s-1)})$
9 $\quad\quad \nu \leftarrow \text{update}_{(s)}(\nu, y^{(j)}, j)$
10 $\quad\quad \mathcal{C}_{(s)}^{j+1} \leftarrow \mathcal{C}_{(s)}^j \cup \{y^{(j)}\}$
11 $\quad\quad j \leftarrow j + 1$
12 $\quad$ **if** $|\mathcal{C}_{(s)}^j| < |\mathcal{C}_{(s-1)}|$ **then**
13 $\quad\quad \nu_i \leftarrow \nu$
14 $\quad\quad m \leftarrow m + 1$
15 $\lambda_{(s)} \leftarrow \lambda^* \left(\{\nu_i\}_{i=1}^{m} ; \alpha_{(s)}\right)$
16 **return** $\lambda_{(s)}$

we perform a grid search over combinations of $\beta_{(1)}$ and $\beta_{(2)}$ that correspond to $\alpha$ and choose the configuration that minimizes admissibility (more details in App. D.3). We do this for each experiment separately and compare the optimal configuration to our method at level $\alpha$.

**Metrics.** 1) # Queries ($\downarrow$): Number of required queries to the admission function during calibration. 2) Time ($\downarrow$): Runtime (in seconds) for an entire calibration, excluding preprocessing (such as splitting data). 3) Set Size ($\downarrow$): Size of the final prediction set. 4) Frac. Reject ($\downarrow$): The fraction of rejected calibrations. For SCOPE-Gen, this can occur if the conformal quantile (6) is not finite (§ 4.2). For CLM, this occurs if the null hypothesis cannot be rejected even once.

**Baselines & Ablations .** 1) CLM (Quach et al., 2024). 2) CLM reduced `max`: This baseline assesses how CLM performs if we enforce a lower amount of admissibility checks during calibration by limiting `max`. Specifically, we choose `max = 10` as limit. 3) SCOPE-Gen gen.-only: We only perform SCOPE-Gen using the generation stage and do not apply any filters. 4) SCOPE-Gen flipped: We apply the quality filter first and the diversity filter second (§ 4.1; config. 1 [App. D.1]; when both filters are used). We present 3) and 4) in App. F.

## 5.1 SETUP

For sake of demonstration, experiments shown here use automated admission functions. We leave experiments that involve admissibility checks made by human domain experts for future work[7]. We provide a general description in the main text and refer to App. E.1 regarding setup and App. E.2 for examples of each task. For SCOPE-Gen, we show config. 1 (see App. D.1) in the main text.

**Natural Language Generation.** For all tasks, the default prediction upper bound is chosen as `max = 20`. We adopt three tasks from Quach et al. (2024) related to natural language generation. 1) TriviaQA (Joshi et al., 2017): Open-domain question answering using LLama-2 (7B) (Touvron et al., 2023), without fine-tuning. An answer $y$ is admissible if it exactly matches one of the ground truth answers $y^t$. 2) MIMIC-CXR (Johnson et al., 2019): Radiology report generation using a pre-trained

---

[7] We stress that admissibility control (1) still applies if a human domain expert assesses admissibility.

Table 1: **Quantitative Evaluations.** All metrics $\pm 1$ standard deviation of the different baselines from 300 repeated evaluations. For each evaluation, we first sub-sample a calibration set of size $n = 600$ and calibrate each method at coverage level $\alpha = 0.3$ with *sum* non-conformity (10). Then, we take means for each metric on a test set of size 300, sampled from the remaining data. Best is bold.

| Method | Metric | TriviaQA | MIMIC-CXR | CNN/DM | Molecules |
|---|---|---|---|---|---|
| CLM | # Queries | $6.570 \pm 0.210$ | $20.000 \pm 0.000$ | $20.000 \pm 0.000$ | $19.280 \pm 0.223$ |
| | Time | $2.710 \pm 0.019$ | $113.677 \pm 10.891$ | $105.907 \pm 13.137$ | $15.853 \pm 1.441$ |
| | Set Size | $2.011 \pm 0.388$ | $14.434 \pm 2.229$ | $6.720 \pm 2.244$ | $17.448 \pm 3.645$ |
| | Frac. Reject | $\mathbf{0.000 \pm 0.000}$ | $0.240 \pm 0.000$ | $\mathbf{0.023 \pm 0.000}$ | $\mathbf{0.097 \pm 0.000}$ |
| CLM reduced `max` | # Queries | $4.187 \pm 0.106$ | $10.000 \pm 0.000$ | $10.000 \pm 0.000$ | $\mathbf{7.109 \pm 0.069}$ |
| | Time | $0.971 \pm 0.012$ | $26.336 \pm 1.615$ | $26.996 \pm 1.354$ | $-$ |
| | Set Size | $2.463 \pm 0.776$ | $16.226 \pm 1.790$ | $6.190 \pm 1.792$ | $-$ |
| | Frac. Reject | $0.020 \pm 0.000$ | $0.817 \pm 0.000$ | $0.207 \pm 0.000$ | $1.000 \pm 0.000$ |
| SCOPE-Gen | # Queries | $\mathbf{2.910 \pm 0.229}$ | $\mathbf{4.214 \pm 1.011}$ | $\mathbf{3.456 \pm 0.478}$ | $8.606 \pm 0.594$ |
| | Time | $\mathbf{0.026 \pm 0.001}$ | $\mathbf{0.072 \pm 0.030}$ | $\mathbf{0.061 \pm 0.013}$ | $\mathbf{0.093 \pm 0.002}$ |
| | Set Size | $\mathbf{1.202 \pm 0.164}$ | $\mathbf{7.390 \pm 1.856}$ | $\mathbf{3.669 \pm 0.910}$ | $\mathbf{9.409 \pm 1.099}$ |
| | Frac. Reject | $\mathbf{0.000 \pm 0.000}$ | $\mathbf{0.203 \pm 0.000}$ | $0.057 \pm 0.000$ | $\mathbf{0.097 \pm 0.000}$ |

vision transformer (Dosovitskiy, 2020) and GPT2-small (Radford et al., 2019). Admissibility is assessed using soft CheXbert labels (Smit et al., 2020). 3) CNN/DM (Hermann et al., 2015): News article summarization. A fine-tuned T5-XL model (Raffel et al., 2020) is used and its outputs are compared to manually generated summaries. Admissibility is assessed via ROUGE-L (Lin, 2004).

**Molecular Graph Extension.** We use DiGress (Vignac et al., 2023), a discrete diffusion model for molecular graph generation that was trained on the MOSES data set (Polykovskiy et al., 2020), which consists of 1,936,962 drug-like compounds. We use a molecule $y^t$ from the calibration set and remove a single atom at random (subject to validity), leading to an "incomplete" compound $x$. We then condition DiGress on $x$ via masking to generate single-atom extensions $y$. An extension $y$ is considered admissible (i.e., $a(y, y^t) = 1$ (2)) if it exactly matches $y^t$. For this experiment, we only apply a quality filter, where the quality function $q$ corresponds to the quantitative estimate of drug-likeness (QED; Bickerton et al. (2012)). The prediction upper bound is chosen as $\texttt{max} = 40$.

## 5.2 RESULTS

SCOPE-Gen consistently outperforms CLM in terms of amount of admissibility checks (# Queries) and runtime (Time), as can be seen in Tab. 1. For instance, on MIMIC-CXR, SCOPE-Gen requires only $\frac{4.21}{20} \approx 21\%$ of the admissibility queries required for CLM. The "CLM reduced `max`" baseline demonstrates that counteracting this disadvantage by reducing the maximum sample size `max` comes at a high price to pay: The amount of rejected calibrations increases greatly (e.g., $81.7\%$ on MIMIC-CXR), while still being far from SCOPE-Gen in terms of performance. This decreased performance is due to the fact that the admissibility guarantee in CLM holds for this specified `max` parameter, also during inference. In the inference phase, however, an upper bound on the prediction set size is often irrelevant, because no more admissibility checks must be performed. In addition, SCOPE-Gen consistently generates smaller prediction sets when the *count* or *sum* non-conformity updates are chosen (see (10)). The only setting where CLM outperforms SCOPE-Gen in terms of prediction set size is for the *max* baseline, but not across all experiments (shown in App. F). An additional benefit of SCOPE-Gen in comparison to CLM is the vast reduction in computational demand for calibration: For MIMIC-CXR, SCOPE-Gen takes on average only 0.072 seconds for a single calibration, in comparison to CLM which takes 113.677 seconds on average ($\frac{0.072}{113.677} \approx 10^{-4} \cdot 6.3$).

## 6 RELATED WORK

Conformal prediction and the more general field known as distribution-free risk control have motivated a plethora of developments related to various aspects such as causality (Lei & Candès, 2021; Schröder et al., 2024), non-exchangeable data (Barber et al., 2023; Angelopoulos et al., 2024; Oliveira et al., 2024), conditional coverage (Vovk, 2012; Lei & Wasserman, 2014; Cauchois et al., 2021; Deutschmann et al., 2023), and many others. The present work is also closely related to *active*

*learning* (Settles, 2009), which employs an *oracle-in-the-loop* framework to reduce the number of queries made to the oracle. To reduce the scope of this section, we proceed to highlight advancements in the applications of risk control in the context of sequential filtering and generative models.

**Risk Control and Sequential Filtering.** Sequential (or iterative/cascading) filtering procedures are ubiquitous in the machine learning literature, as demonstrated by, e.g., Deng & Rush (2020); Lahmiri et al. (2021); Choi & Yang (2022). Among these, the method that most closely aligns with our approach is that of Fisch et al. (2021). Similarly to the present work, Fisch et al. (2021) sequentially filter a set of candidate solutions in order to obtain a prediction set that controls admissibility (1). In contrast to the present work, Fisch et al. (2021) consider a classification scenario where filtering is done via increasingly complex classifiers to an initially assumed admissible set of candidates. This assumption contrasts with our setting, where we derive the initial set from a generative model without assuming admissibility. Moreover, Fisch et al. (2021) treat each candidate solution independently, disregarding the order of sampling within each filter stage, which contrasts with our greedy filters (§ 4.1). This is a critical limitation when pruning prediction sets from generative models, as Fisch et al. (2021) cannot take similarities between outputs into account.

**Risk Control for Generative Models.** One line of work constructs confidence intervals for generative models in the pixel space of images (Horwitz & Hoshen, 2022; Teneggi et al., 2023). These studies focus on image-to-image tasks, such as denoising, where upper and lower uncertainty bounds of conformal prediction intervals on the target image can be visualized well. Conformal prediction has also been applied to planning with diffusion models by utilizing conformal prediction for the reward estimate (Sun et al., 2024). Other applications include density estimation (Diamant et al., 2024) and detection of adversarial examples (Cai et al., 2020). Wang et al. (2023) use generative models to create conformal prediction sets by determining a radius of a ball around each sample such that the union over all balls controls coverage. In contrast to our approach, the method proposed by Wang et al. (2023) is not suitable for non-Euclidean or high-dimensional data. Another work uses normalizing flows to approximate conditional coverage for regression (Colombo, 2024).

Works other than the one by Quach et al. (2024) that target language generation with a different scope to ours include the approach by Ulmer et al. (2024) who consider language modeling for non-exchangeable data, Gui et al. (2024) who focus on conformal alignment of language models and Kumar et al. (2023) who specifically consider multiple choice problems. Another concurrent line of work, followed by Cherian et al. (2024); Mohri & Hashimoto (2024); Liu & Wu (2024), provides guarantees on individual long-form text outputs by chunking them into sub-claims and removing false sub-claims. A critical assumption for such methods to be practical is that text responses indeed consist of multiple distinct claims with little coherence.

**Detecting Hallucination without Guarantees.**   There is a body of work focused on quantifying uncertainty and detecting hallucinations in language models, though these approaches do not directly achieve distribution-free risk control. Existing methods in this domain are based on notions such as semantic (Kuhn et al., 2023; Lin et al., 2024) or task-specific uncertainty (Wang & Holmes, 2024) and self-consistency (Zhang et al., 2024a). Although not explored in the present work, we believe that such approaches offer a promising opportunity for cross-pollination with ideas from conformal prediction in future work.

## 7    CONCLUSION

In the present work, we introduced SCOPE-Gen, a sample-efficient conformal prediction method for generative models. This method sequentially prunes an initial prediction set from a generative model using greedy filters, allowing the total admissibility to factorize as a Markov chain. Consequently, each pruning step can be calibrated separately. Experimentally, SCOPE-Gen has shown to require fewer queries of an admissibility function to maintain control over the total admissibility at a desired level, compared to prior work. This efficiency is particularly valuable in domains where admissibility assessments are costly, such as those requiring evaluations by human domain experts. We hope that our work inspires further research in risk control for generative models, with a focus on integrating application-specific constraints and desiderata, such as sample efficiency.

## REPRODUCIBILITY STATEMENT

Our source code is available on GitHub at https://github.com/rudolfwilliam/scope-gen. Instructions for reproducibility are provided in the `README.md` file at the top level of the code directory. All parameters for running the experiments can be found in the `config` folders within the `script` folders of each experiment folder.

## ACKNOWLEDGEMENTS

Michael Muehlebach is funded by the German Research Foundation (DFG) under the project 456587626 of the Emmy Noether Programme. The authors thank the International Max Planck Research School for Intelligent Systems (IMPRS-IS) for supporting Klaus-Rudolf Kladny.

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

# Appendix

## Table of Contents

## A   RELATIONSHIP BETWEEN RISK CONTROL AND ADMISSIBILITY CONTROL

Conformal risk control (Angelopoulos et al., 2022) considers the generation of a prediction set $\mathcal{C}_{\boldsymbol{\lambda}}$ with calibration parameters $\boldsymbol{\lambda}$ computed from a labeled i.i.d. calibration set $\{(X_i, Y_i)\}_{i=1}^n$, a new input/condition $X_{n+1}$ with label $Y_{n+1}$ and a loss function $l$. The desired property of interest for $\mathcal{C}_{\boldsymbol{\lambda}}$ is called *risk control*, defined as

$$\mathbb{E}[l(\mathcal{C}_{\boldsymbol{\lambda}}(X_{n+1}), Y_{n+1}^t)] < \alpha, \tag{13}$$

where $\alpha > 0$. A classic example is the coverage loss $l(\mathcal{C}_{\boldsymbol{\lambda}}(x), y^t) = \mathbb{1}\{y^t \notin \mathcal{C}_{\boldsymbol{\lambda}}(x)\}$ (Vovk et al., 2005), which gives (13) the simple interpretation that $\mathcal{C}_{\boldsymbol{\lambda}}(X_{n+1})$ contains $Y_{n+1}$ with probability $\geq 1 - \alpha$. In the present work, we consider the loss

$$l(\mathcal{C}_{\boldsymbol{\lambda}}(x), y^t) = \mathbb{1}\{\nexists y \in \mathcal{C}_{\boldsymbol{\lambda}}(x) : a(y, y^t) = 1\}. \tag{14}$$

When inserting (14) into (13), our definition of *admissibility control* (1) arises naturally:

$$\mathbb{E}[\mathbb{1}\{\nexists Y \in \mathcal{C}_{\boldsymbol{\lambda}}(X_{n+1}) : a(Y, Y_{n+1}^t) = 1\}] < \alpha$$
$$\iff \mathbb{E}[\mathbb{1}\{\exists Y \in \mathcal{C}_{\boldsymbol{\lambda}}(X_{n+1}) : a(Y, Y_{n+1}^t) = 1\}] \geq 1 - \alpha$$
$$\iff \mathbb{P}(\exists Y \in \mathcal{C}_{\boldsymbol{\lambda}}(X_{n+1}) : a(Y, Y_{n+1}^t) = 1) \geq 1 - \alpha.$$

## B   DEFINITION OF THE EMPIRICAL QUANTILE

We define the empirical $\delta$-quantile for $\delta \in \mathbb{R}$ at 1-dimensional real-valued data points $\{x_i\}_{i=1}^n$ as

$$\widehat{\text{quantile}}\left(\{x_i\}_{i=1}^n;\ \delta\right) := \inf_{q \in \mathbb{R} \cup \{\infty\}} \left\{q : \sum_{i=1}^n \mathbb{1}\{x_i \leq q\} \geq n\delta\right\}. \tag{15}$$

Intuitively, (15) corresponds to the $n\delta^{\text{th}}$ smallest element in $\{x_i\}_{i=1}^n$ if $\delta \leq 1$. Otherwise ($\delta > 1$), the empirical quantile (15) is set to $\infty$.

## C   BOUNDING CONFORMAL ADMISSIBILITY BY LEARN-THEN-TEST ADMISSIBILITY

*Proof.* Let $\beta_{(1)} \in (0, 1)$ and $\beta_{(2)} \in (0, 1)$ be chosen arbitrarily, such that $\beta_{(1)} + \beta_{(2)} - \beta_{(1)}\beta_{(2)} = \alpha$. We further suppose that the learn-then-test guarantee

$$\mathbb{P}\left[\mathbb{P}\left(A(\mathcal{C}_{\boldsymbol{\lambda}}) = 1 \,|\, D_{\text{cal}}\right) \geq 1 - \beta_{(1)}\right] \geq 1 - \beta_{(2)} \tag{16}$$

is satisfied. We now show that the admissibility is bounded from below as

$$\mathbb{E}[\mathbb{1}\{A(\mathcal{C}_{\boldsymbol{\lambda}}) = 1\}] \geq 1 - \alpha.$$

To this end, we observe that the admissibility

$$\mathbb{E}\left[\mathbb{1}\{A(\mathcal{C}_{\boldsymbol{\lambda}}) = 1\}\right] = \mathbb{E}\left[\mathbb{P}\left[A(\mathcal{C}_{\boldsymbol{\lambda}}) = 1 \,|\, D_{\text{cal}}\right]\right]$$

can be split into two parts

$$\mathbb{E}\left[\mathbb{1}\{\mathbb{P}\left[A(\mathcal{C}_{\boldsymbol{\lambda}}) = 1 | D_{\text{cal}}\right] \geq 1 - \beta_{(1)}\}\mathbb{P}\left[A(\mathcal{C}_{\boldsymbol{\lambda}}) = 1 \,|\, D_{\text{cal}}\right]\right] + \tag{17}$$
$$\mathbb{E}\left[\mathbb{1}\{\mathbb{P}\left[A(\mathcal{C}_{\boldsymbol{\lambda}}) = 1 | D_{\text{cal}}\right] < 1 - \beta_{(1)}\}\mathbb{P}\left[A(\mathcal{C}_{\boldsymbol{\lambda}}) = 1 \,|\, D_{\text{cal}}\right]\right]. \tag{18}$$

We note that (18) is bounded from below by 0. We further bound (17) as follows

$$\mathbb{E}\left[\mathbb{1}\{\mathbb{P}\left[A(\mathcal{C}_{\boldsymbol{\lambda}}) = 1 | D_{\text{cal}}\right] \geq 1 - \beta_{(1)}\}\mathbb{P}\left[A(\mathcal{C}_{\boldsymbol{\lambda}}) = 1 \,|\, D_{\text{cal}}\right]\right]$$
$$\geq (1 - \beta_{(1)}) \cdot \mathbb{P}\left[\mathbb{P}\left[A(\mathcal{C}_{\boldsymbol{\lambda}}) = 1 \,|\, D_{\text{cal}}\right] \geq 1 - \beta_{(1)}\right]$$
$$\geq (1 - \beta_{(1)}) \cdot (1 - \beta_{(2)}),$$

where the last inequality holds by the learn-then-test guarantee (16). $\qquad \square$

## D  IMPLEMENTATION DETAILS

### D.1  PARAMETER CONFIGURATION

The most straightforward choice is to equally distribute the risk among prediction steps, resulting in

$$\alpha_{(s)} = 1 - (1 - \alpha)^{1/K}, \tag{19}$$

where $K$ is the amount of prediction steps (typically, $K = 3$). We note that (19) is somewhat analogous to a Bonferroni correction in that it also equally distributes a correction. In practice, when working with a upper bound on the sample size `max` (outlined in § 4.2), however, lower admissibility levels can often be attained by different choice of $\alpha_{(s)}$. Specifically, we choose a larger risk level for the generation step ($s = 0$) and distribute the remaining risk equally among the other step(s) ($s \in \{1, 2\}$). We assess two different configurations of SCOPE-Gen, which we name config. 1 and config 2. For config. 1, we choose $M = 5$ and set

$$\alpha_{(0)} = 1 - (1 - \alpha)^{(M-1)/M}$$

and

$$\alpha_{(s)} = 1 - (1 - \alpha)^{1/(M(K-1))}, \ \forall s > 0.$$

For config. 2 , we use the same setup as for config. 1, with the only difference that we only use the diversity filter (i. e., $K = 2$). In addition, we add a prediction set count penalty to the diversity update function, resulting in

$$\texttt{update}(\nu, y) = \begin{cases} -d_{\max}, & \text{if} \quad \nu = 0 \\ - \min_{y' \in \mathcal{C}_{(s)} \backslash \{y\}} d(y, y') + \gamma_d |\mathcal{C}_{(s)}|, & \text{else,} \end{cases}$$

with $\gamma_d = 0.1$.

### D.2  DISTANCE UPPER BOUND

All distance functions in the present work are derived from text-based similarity functions such as `ROUGE` (Lin, 2004). Such metrics typically range from 0 to 1 and we can convert them to distance functions simply by negating them (such that they range from $-1$ to 0). Thus, for such metrics, we obtain a distance lower bound by choosing $d_{\max} = -1$. If $d$ is not bounded, we can turn it into a bounded distance $\tilde{d}$, for example by taking

$$\tilde{d}(y, y') := -\frac{1}{1 + d(y, y')},$$

assuming positivity of $d$.

### D.3  GRID SPECIFICATION FOR CLM

In general, we observe empirically that coverage is minimized when $\beta_{(2)}$ is much smaller than $\beta_{(1)}$ (usually for $\beta_{(2)} \approx \beta_{(1)}/10$). For a given $\alpha$, we set the grid over $\beta_{(2)}$, denoted as $\boldsymbol{\beta}_{(2)}$, by selecting 10 equidistant points between $\alpha/15$ and $\alpha/5$. This can be mathematically represented as follows:

$$\boldsymbol{\beta}_{(2)} = \left[ \alpha/15 + i \cdot \frac{\alpha/5 - \alpha/15}{9} : i = 0, 1, \ldots, 9 \right].$$

The corresponding values $\boldsymbol{\beta}_{(1)}$, accordingly are

$$\boldsymbol{\beta}_{(1)} = \left[ \frac{1 - \alpha - \boldsymbol{\beta}_{(2)}^{(i)}}{1 - \boldsymbol{\beta}_{(2)}^{(i)}} : i = 0, 1, \ldots, 9 \right].$$

# E    EXPERIMENTAL DETAILS

## E.1    SETUP DETAILS

**Natural Language Generation.**    All of our experiments regarding natural language generation roughly follow the setup from (Quach et al., 2024). Thus, we only highlight differences to the setup considered there:

TriviaQA. We remove duplicates automatically rather than calibrating a parameter to do this, in order to improve statistical efficiency of both methods. For CLM, we set the diversity parameter $\lambda_{(2)} = 0.5$ such that all duplicates are removed and for SCOPE-Gen, we use a fixed diversity filter at the end of the prediction pipeline, which removes duplicates.

MIMIC-CXR. We slightly modify the admissibility criterion due to insufficient implementation details available. Following the approach by Quach et al. (2024), which uses exact CheXbert prediction matches as labels, we obtain few admissible examples ($\approx 30\%$). To address this problem, we adjust the setup by extracting soft labels from the CheXbert model (using weights from the original work). Specifically, we employ a labeling function based on the F1 score

$$\text{F1} := \frac{2}{\text{recall}^{-1} + \text{precision}^{-1}}, \tag{20}$$

where both precision and recall are computed from 14 clinically relevant labels output by the CheXbert model. This F1 score aligns with the evaluation metric used in the original work by Smit et al. (2020). We consider a prediction admissible if $\text{F1} > 0.6$, a manually chosen threshold. We compare all baselines using the predictions of this model.

**Molecular Graph Extension.**    The weights of the used DiGress model can be found on the official GitHub page. In order to condition the generated molecular graphs on a given substructure, we employ a form of masking in the reverse diffusion process (from noise to data distribution). This is similar to masking approaches in computer vision, such as the one proposed by Lugmayr et al. (2022). However, unlike Lugmayr et al. (2022), we mask the given graph structure directly (rather than applying noise to it) and do not repeat the reverse diffusion process multiple times.

Analagously to TriviaQA, we filter out duplicates automatically. In addition, we treat invalid predictions by setting their quality estimate to $0$. Invalid molecules are molecules that cannot be represented in a valid resonance form according to Kekulé's rules.

In order to check whether two molecules are identical (criterion for admissibility), we convert both molecules to their canonical SMILES representations and check whether these are identical.

## E.2    EXAMPLES

We demonstrate one specific example per experiment, where we show both condition $x$ and ground truth output $y^t$.

**TriviaQA:** The condition $x$ is a question in natural language, the ground truth target $y^t$ is a short phrase in natural language (with multiple aliases that are all considered admissible).

|  |  |
|---|---|
| "Which Lloyd Webber musical premiered in the US on 10th December 1993?" | "Sunset Blvd", "West Sunset Boulevard", "Sunset Boulevard", "Sunset Bulevard", "Sunset Blvd." |
| $x$ | $y^t$ |

**CNN/DM:** The condition $x$ is a lengthy news report in natural language, the ground truth target $y^t$ is a manually generated summary (in natural language) of $x$.

" having been on the receiving end of a 6-1 thumping , a defeat like that could be justifiably met with a backlash by angry supporters . watching a 3-1 first leg aggregate advantage turn into a 7-4 deficit come the end of the reverse encounter too could send many fans apoplectic at the capitulation of their side . [...] . it was the first time that porto , who had been unbeaten in this season 's tournament up until tuesday night , had reached the quarter-finals of the champions league since the 2008-09 season . "

$$x$$

bayern munich beat porto 6-1 in their champions league tie on tuesday . result saw bayern win quarter-final encounter 7-4 on aggregate . it was the first-time porto had reached that stage since the 2008-09 season .

$$y^t$$

**MIMIC-CXR:** The condition $x$ is a anterior to posterior or posterior to anterior chest X-ray image (recorded from the front or from the back, but not from the side), including a text prompt in natural language, which contains all text prior to either the "FINDINGS", "IMPRESSION" or "FINDINGS AND IMPRESSION" keyword of the ground truth report. The ground truth target $y^t$ is all text starting from the identified keyword.

FINAL REPORT
HISTORY: ___-year-old man with fever, rule out pneumonia.
TECHNIQUE: Portable erect chest radiograph was obtained.
COMPARISON: Chest radiograph from ___.

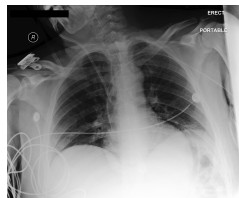

FINDINGS: Right central venous line is in stable position at the mid SVC, and the left PICC line ends near the cavoatrial junction. Gastric tube passes below the diaphragm and ends in the body of the stomach. Low lung volumes continue to be seen. Previous vascular congestion has improved, and the heart size and mediastinal contours are normal.
IMPRESSION: No focal consolidation to suggest pneumonia.

$$x \qquad\qquad\qquad y^t$$

**Molecules:** The condition $x$ is given as a valid scaffold (i.e. a subgraph that adheres to the grammar of valid molecules) that results as a random removal of a single atom (including all of its connections; highlighted in cyan) from a compound $y^t$ in the calibration set.

$$x \qquad\qquad\qquad y^t$$

## F    ADDITIONAL EXPERIMENTS

### F.1    ADMISSIBILITY ANALYSIS

We demonstrate the empirical admissibility levels for the MIMIC-CXR experiment at admissibility level $\alpha = 0.3$, varying calibration set size $n$ and the used non-conformity measure. We use a cross-validation like algorithm (Angelopoulos & Bates (2021); Fig. 12) to generate histograms. The rest of the experimental setup is identical to the setup used for all other experiments (see caption of Tab. 1)

As shown in the histograms of Fig. 3, for all non-conformity measures, the admissibility achieved by SCOPE-Gen (config. 1 [App. D.1]; blue) is slightly conservative for low sample sizes and converges to the desired level as the calibration set size $n$ is increased from 300 to 1200. This is similar for CLM (orange), where the achieved admissibility (using the optimization strategy described in § 5) also becomes less conservative for large sample sizes. However, in contrast to SCOPE-Gen, CLM is

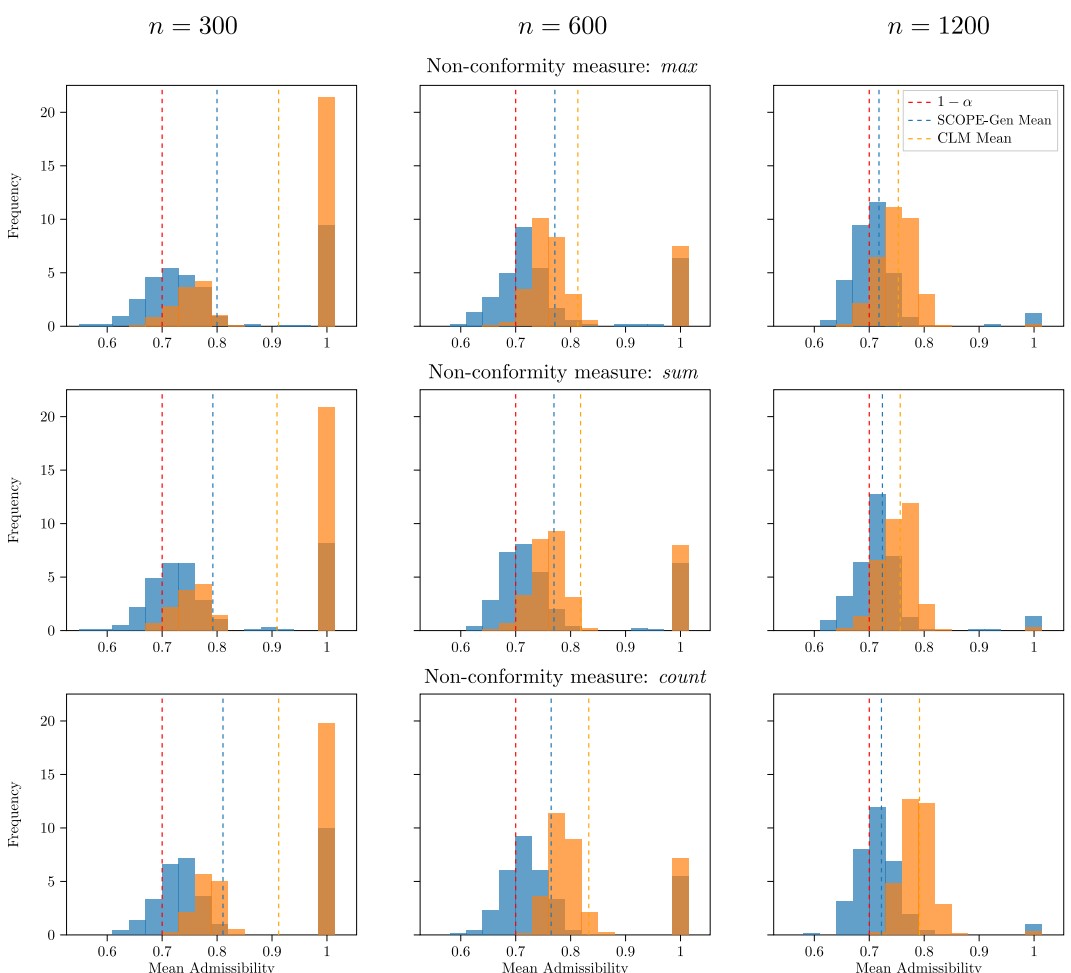

Figure 3: **Admissibility Analysis for MIMIC-CXR.** For all non-conformity scores, SCOPE-Gen (config. 1, [see App. D.1]; blue) becomes less conservative with respect to the desired admissibility level $1 - \alpha$ (red dashed line) as the amount of calibration samples $n$ increases, as typical for methods based on conformal prediction. For CLM (orange), in contrast, conservativeness depends much on the chosen non-conformity measure.

generally more conservative. In addition, this conservativeness is highly influenced by the choice of non-conformity measure: While the CLM gap becomes tighter for the *max* and *sum* non-conformities, this is not the case for the *count* non-conformity, where a large gap persists even for $n = 1200$.

A further observation is that SCOPE-Gen tends to reject far fewer calibration sets at small calibration set size ($n = 300$). This can be seen from the bars at coverage $1.$, which indicate calibration sets that are rejected (rejecting a calibration set amounts to returning the entire output space $\mathcal{Y}$ (see § 2), which results in a $100\%$ admissibility). We see that the blue bar (SCOPE-Gen) tends to be smaller than the orange bar (CLM).

### F.2  FURTHER QUANTITATIVE EVALUATIONS  & ABLATIONS

We demonstrate further experiments that showcase the performance for the different non-conformity measures (*count* and *max* ($\gamma = 0.1$) in addition to *sum* ($\gamma = 0.5$); see (10)), including two more baselines (baselines 3) and 4) introduced in § 5) for admissibility levels $\alpha \in \{0.3, 0.35, 0.4\}$ and both configurations of SCOPE-Gen (see App. D.1). While for some experiments, choosing smaller values for $\alpha$ is feasible (e.g., on TriviaQA, as can be seen in App. G.1), this is not possible for all experiments due to the performance of the underlying model. Specifically, if a model does not generate a single admissible example within the upper limit max for more than $\alpha$ fraction of examples,

we see that generating prediction sets that fulfill an admissibility of level smaller than $\alpha$ cannot be possible, irrespective of the calibration method that is used. We observe such issues for CNN/DM and Molecules when using admissibility levels smaller than $\alpha = 0.3$. Thus, we restrict our numerical experiments to larger admissibility levels.

We see in Tab. 2 that typically either SCOPE-Gen (config. 1 and config. 2, see App. D.1) or one of its ablations performs best in terms of the considered evaluation metrics. We see that occasionally, SCOPE-Gen gen. only generates the smallest prediction sets on average. We hypothesize that this may have to do with the fact that SCOPE-Gen gen. will tend to be slightly less affected by variance in the estimation of the threshold, since SCOPE-Gen gen. has more data to estimate the latter. In terms of the non-conformity update function, we see that a general recommendation in favor of one over any other cannot be made. For example, considering the setting $\alpha = 0.3$ for TriviaQA, we see that SCOPE-Gen config. 1 generates the smallest prediction sets for the *sum* non-conformity, but the fewest admissibility checks are achieved for the same method when using the *count* non-conformity. For $\alpha = 0.3$ and MIMIX-CXR, SCOPE-Gen config. 2 generates the smallest prediction sets, using *max* non-conformity.

Table 2: **Further Quantitative Evaluations.** See Tab. 1 for experimental setup. Best is bold.

| Method | Metric | TriviaQA | MIMIC-CXR | CNN/DM | Molecules |
|---|---|---|---|---|---|
| | | | $\alpha = 0.3$ | | |
| | | | Non-Conformity: *sum* | | |
| CLM | # Queries | $6.570 \pm 0.210$ | $20.000 \pm 0.000$ | $20.000 \pm 0.000$ | $19.280 \pm 0.223$ |
| | Time | $2.710 \pm 0.019$ | $113.677 \pm 10.891$ | $105.907 \pm 13.137$ | $15.853 \pm 1.441$ |
| | Set Size | $2.011 \pm 0.388$ | $14.434 \pm 2.229$ | $6.720 \pm 2.244$ | $17.448 \pm 3.645$ |
| | Frac. Reject | $\mathbf{0.000 \pm 0.000}$ | $0.240 \pm 0.000$ | $0.023 \pm 0.000$ | $0.097 \pm 0.000$ |
| CLM reduced max | # Queries | $4.187 \pm 0.106$ | $10.000 \pm 0.000$ | $10.000 \pm 0.000$ | $\mathbf{7.109 \pm 0.069}$ |
| | Time | $0.971 \pm 0.012$ | $26.336 \pm 1.615$ | $26.996 \pm 1.354$ | $-$ |
| | Set Size | $2.463 \pm 0.776$ | $16.226 \pm 1.790$ | $6.190 \pm 1.792$ | $-$ |
| | Frac. Reject | $0.020 \pm 0.000$ | $0.817 \pm 0.000$ | $0.207 \pm 0.000$ | $1.000 \pm 0.000$ |
| SCOPE-Gen config. 1 | # Queries | $\mathbf{2.910 \pm 0.229}$ | $\mathbf{4.214 \pm 1.011}$ | $\mathbf{3.456 \pm 0.478}$ | $8.606 \pm 0.594$ |
| | Time | $0.026 \pm 0.001$ | $0.072 \pm 0.030$ | $0.061 \pm 0.013$ | $0.093 \pm 0.002$ |
| | Set Size | $\mathbf{1.202 \pm 0.164}$ | $7.390 \pm 1.856$ | $3.669 \pm 0.910$ | $\mathbf{9.409 \pm 1.099}$ |
| | Frac. Reject | $\mathbf{0.000 \pm 0.000}$ | $0.203 \pm 0.000$ | $0.057 \pm 0.000$ | $0.097 \pm 0.000$ |
| SCOPE-Gen config. 2 | # Queries | $-$ | $5.216 \pm 0.710$ | $4.128 \pm 0.452$ | $-$ |
| | Time | $-$ | $0.023 \pm 0.007$ | $0.017 \pm 0.003$ | $-$ |
| | Set Size | $-$ | $\mathbf{5.839 \pm 1.144}$ | $\mathbf{2.973 \pm 0.614}$ | $-$ |
| | Frac. Reject | $-$ | $0.090 \pm 0.000$ | $0.017 \pm 0.000$ | $-$ |
| SCOPE-Gen gen. only | # Queries | $3.611 \pm 0.166$ | $6.518 \pm 0.310$ | $5.525 \pm 0.297$ | $9.888 \pm 0.412$ |
| | Time | $\mathbf{0.005 \pm 0.000}$ | $\mathbf{0.004 \pm 0.000}$ | $\mathbf{0.004 \pm 0.001}$ | $\mathbf{0.005 \pm 0.000}$ |
| | Set Size | $2.050 \pm 0.184$ | $6.003 \pm 1.040$ | $3.302 \pm 0.579$ | $10.038 \pm 0.984$ |
| | Frac. Reject | $\mathbf{0.000 \pm 0.000}$ | $\mathbf{0.000 \pm 0.000}$ | $\mathbf{0.000 \pm 0.000}$ | $\mathbf{0.000 \pm 0.000}$ |
| SCOPE-Gen flipped | # Queries | $-$ | $4.581 \pm 1.121$ | $3.530 \pm 0.531$ | $-$ |
| | Time | $-$ | $0.048 \pm 0.013$ | $0.044 \pm 0.005$ | $-$ |
| | Set Size | $-$ | $7.848 \pm 1.976$ | $3.635 \pm 0.924$ | $-$ |
| | Frac. Reject | $-$ | $0.193 \pm 0.000$ | $0.033 \pm 0.000$ | $-$ |
| | | | Non-Conformity: *count* | | |
| CLM | # Queries | $6.581 \pm 0.210$ | $20.000 \pm 0.000$ | $20.000 \pm 0.000$ | $19.263 \pm 0.217$ |
| | Time | $2.260 \pm 0.015$ | $90.593 \pm 7.089$ | $84.864 \pm 9.882$ | $16.118 \pm 3.096$ |
| | Set Size | $3.912 \pm 1.104$ | $18.221 \pm 1.616$ | $15.762 \pm 3.265$ | $23.209 \pm 3.153$ |
| | Frac. Reject | $\mathbf{0.000 \pm 0.000}$ | $0.213 \pm 0.000$ | $0.020 \pm 0.000$ | $0.087 \pm 0.000$ |
| CLM reduced max | # Queries | $4.203 \pm 0.109$ | $10.000 \pm 0.000$ | $10.000 \pm 0.000$ | $\mathbf{7.117 \pm 0.067}$ |
| | Time | $0.465 \pm 0.007$ | $10.922 \pm 0.325$ | $10.011 \pm 0.332$ | $-$ |
| | Set Size | $3.660 \pm 0.880$ | $17.822 \pm 1.462$ | $12.530 \pm 3.353$ | $-$ |
| | Frac. Reject | $0.043 \pm 0.000$ | $0.800 \pm 0.000$ | $0.190 \pm 0.000$ | $1.000 \pm 0.000$ |
| SCOPE-Gen config. 1 | # Queries | $\mathbf{2.879 \pm 0.223}$ | $\mathbf{4.466 \pm 1.038}$ | $\mathbf{3.514 \pm 0.489}$ | $8.530 \pm 0.584$ |
| | Time | $0.010 \pm 0.000$ | $0.059 \pm 0.033$ | $0.044 \pm 0.016$ | $0.012 \pm 0.001$ |
| | Set Size | $\mathbf{1.255 \pm 0.169}$ | $7.866 \pm 1.909$ | $3.850 \pm 1.014$ | $\mathbf{9.853 \pm 1.268}$ |
| | Frac. Reject | $\mathbf{0.000 \pm 0.000}$ | $0.163 \pm 0.000$ | $0.027 \pm 0.000$ | $\mathbf{0.000 \pm 0.000}$ |
| SCOPE-Gen config. 2 | # Queries | $-$ | $5.385 \pm 0.732$ | $4.162 \pm 0.458$ | $-$ |
| | Time | $-$ | $0.015 \pm 0.007$ | $0.010 \pm 0.003$ | $-$ |
| | Set Size | $-$ | $\mathbf{5.899 \pm 1.044}$ | $\mathbf{2.970 \pm 0.551}$ | $-$ |
| | Frac. Reject | $-$ | $0.080 \pm 0.000$ | $0.013 \pm 0.000$ | $-$ |

Continued on next page

| Method | Metric | TriviaQA | MIMIC-CXR | CNN/DM | Molecules |
|---|---|---|---|---|---|
| SCOPE-Gen gen. only | # Queries
Time
Set Size
Frac. Reject | $3.611 \pm 0.166$
$\mathbf{0.004 \pm 0.000}$
$2.106 \pm 0.236$
$\mathbf{0.000 \pm 0.000}$ | $6.518 \pm 0.310$
$\mathbf{0.004 \pm 0.000}$
$6.693 \pm 1.083$
$\mathbf{0.000 \pm 0.000}$ | $5.525 \pm 0.297$
$\mathbf{0.004 \pm 0.001}$
$3.927 \pm 0.731$
$\mathbf{0.000 \pm 0.000}$ | $9.888 \pm 0.412$
$\mathbf{0.004 \pm 0.001}$
$10.049 \pm 0.969$
$\mathbf{0.000 \pm 0.000}$ |
| SCOPE-Gen flipped | # Queries
Time
Set Size
Frac. Reject | –
–
–
– | $4.760 \pm 1.160$
$0.029 \pm 0.013$
$7.744 \pm 2.025$
$0.143 \pm 0.000$ | $3.670 \pm 0.570$
$0.025 \pm 0.006$
$3.945 \pm 0.995$
$0.023 \pm 0.000$ | –
–
–
– |
| Non-Conformity: *max* | | | | | |
| CLM | # Queries
Time
Set Size
Frac. Reject | $6.567 \pm 0.196$
$2.782 \pm 0.024$
$1.787 \pm 0.330$
$\mathbf{0.000 \pm 0.000}$ | $20.000 \pm 0.000$
$113.662 \pm 13.219$
$14.143 \pm 1.998$
$0.223 \pm 0.000$ | $20.000 \pm 0.000$
$106.804 \pm 11.690$
$\mathbf{2.751 \pm 0.596}$
$0.027 \pm 0.000$ | $19.286 \pm 0.230$
$15.086 \pm 0.972$
$\mathbf{2.264 \pm 1.258}$
$0.080 \pm 0.000$ |
| CLM reduced max | # Queries
Time
Set Size
Frac. Reject | $4.209 \pm 0.117$
$0.975 \pm 0.010$
$2.212 \pm 0.662$
$0.033 \pm 0.000$ | $10.000 \pm 0.000$
$26.193 \pm 1.702$
$15.756 \pm 2.014$
$0.823 \pm 0.000$ | $10.000 \pm 0.000$
$26.740 \pm 1.540$
$3.192 \pm 0.692$
$0.223 \pm 0.000$ | $\mathbf{7.108 \pm 0.070}$
–
–
$1.000 \pm 0.000$ |
| SCOPE-Gen config. 1 | # Queries
Time
Set Size
Frac. Reject | $\mathbf{3.054 \pm 0.171}$
$0.024 \pm 0.001$
$\mathbf{1.351 \pm 0.143}$
$\mathbf{0.000 \pm 0.000}$ | $\mathbf{4.176 \pm 1.006}$
$0.071 \pm 0.031$
$7.525 \pm 1.922$
$0.230 \pm 0.000$ | $\mathbf{3.478 \pm 0.474}$
$0.059 \pm 0.013$
$3.892 \pm 0.960$
$0.043 \pm 0.000$ | $8.510 \pm 0.590$
$0.076 \pm 0.002$
$9.720 \pm 1.283$
$\mathbf{0.000 \pm 0.000}$ |
| SCOPE-Gen config. 2 | # Queries
Time
Set Size
Frac. Reject | –
–
–
– | $5.096 \pm 0.744$
$0.021 \pm 0.007$
$\mathbf{5.761 \pm 0.980}$
$0.160 \pm 0.000$ | $4.199 \pm 0.422$
$0.017 \pm 0.002$
$3.078 \pm 0.575$
$0.007 \pm 0.000$ | –
–
–
– |
| SCOPE-Gen gen. only | # Queries
Time
Set Size
Frac. Reject | $3.611 \pm 0.166$
$\mathbf{0.005 \pm 0.000}$
$3.061 \pm 0.142$
$\mathbf{0.000 \pm 0.000}$ | $6.518 \pm 0.310$
$\mathbf{0.004 \pm 0.000}$
$6.549 \pm 0.987$
$\mathbf{0.000 \pm 0.000}$ | $5.525 \pm 0.297$
$\mathbf{0.004 \pm 0.000}$
$3.872 \pm 0.497$
$\mathbf{0.000 \pm 0.000}$ | $9.888 \pm 0.412$
$\mathbf{0.005 \pm 0.000}$
$9.829 \pm 0.964$
$\mathbf{0.000 \pm 0.000}$ |
| SCOPE-Gen flipped | # Queries
Time
Set Size
Frac. Reject | –
–
–
– | $4.395 \pm 1.133$
$0.044 \pm 0.013$
$7.460 \pm 1.866$
$0.220 \pm 0.000$ | $3.629 \pm 0.539$
$0.042 \pm 0.005$
$3.955 \pm 0.988$
$0.033 \pm 0.000$ | –
–
–
– |
| $\alpha = 0.35$ | | | | | |
| Non-Conformity: *sum* | | | | | |
| CLM | # Queries
Time
Set Size
Frac. Reject | $6.581 \pm 0.210$
$2.755 \pm 0.028$
$1.674 \pm 0.224$
$\mathbf{0.000 \pm 0.000}$ | $20.000 \pm 0.000$
$106.192 \pm 14.696$
$11.240 \pm 2.075$
$0.017 \pm 0.000$ | $20.000 \pm 0.000$
$102.641 \pm 12.754$
$4.094 \pm 1.118$
$\mathbf{0.000 \pm 0.000}$ | $19.269 \pm 0.222$
$15.574 \pm 1.669$
$13.573 \pm 2.615$
$\mathbf{0.000 \pm 0.000}$ |
| CLM reduced max | # Queries
Time
Set Size
Frac. Reject | $4.212 \pm 0.114$
$0.977 \pm 0.009$
$1.633 \pm 0.443$
$\mathbf{0.000 \pm 0.000}$ | $10.000 \pm 0.000$
$23.966 \pm 0.578$
$13.774 \pm 2.340$
$0.187 \pm 0.000$ | $10.000 \pm 0.000$
$26.360 \pm 1.630$
$4.245 \pm 1.165$
$\mathbf{0.000 \pm 0.000}$ | $\mathbf{7.107 \pm 0.064}$
$0.874 \pm 0.012$
$9.280 \pm 0.000$
$0.997 \pm 0.000$ |
| SCOPE-Gen config. 1 | # Queries
Time
Set Size
Frac. Reject | $\mathbf{2.757 \pm 0.151}$
$0.026 \pm 0.001$
$\mathbf{1.072 \pm 0.069}$
$\mathbf{0.000 \pm 0.000}$ | $\mathbf{3.881 \pm 0.213}$
$0.058 \pm 0.005$
$4.345 \pm 0.802$
$\mathbf{0.000 \pm 0.000}$ | $3.352 \pm 0.299$
$0.055 \pm 0.005$
$2.904 \pm 0.553$
$\mathbf{0.000 \pm 0.000}$ | $7.973 \pm 0.536$
$0.094 \pm 0.003$
$7.468 \pm 0.432$
$\mathbf{0.000 \pm 0.000}$ |
| SCOPE-Gen config. 2 | # Queries
Time
Set Size
Frac. Reject | –
–
–
– | $4.929 \pm 0.436$
$0.019 \pm 0.002$
$4.190 \pm 0.707$
$0.003 \pm 0.000$ | $3.835 \pm 0.332$
$0.016 \pm 0.001$
$\mathbf{2.083 \pm 0.354}$
$\mathbf{0.000 \pm 0.000}$ | –
–
–
– |
| SCOPE-Gen gen. only | # Queries
Time
Set Size
Frac. Reject | $3.633 \pm 0.036$
$\mathbf{0.006 \pm 0.000}$
$1.685 \pm 0.010$
$\mathbf{0.000 \pm 0.000}$ | $6.518 \pm 0.310$
$\mathbf{0.002 \pm 0.000}$
$\mathbf{3.979 \pm 0.557}$
$\mathbf{0.000 \pm 0.000}$ | $5.525 \pm 0.297$
$\mathbf{0.002 \pm 0.000}$
$2.163 \pm 0.315$
$\mathbf{0.000 \pm 0.000}$ | $9.641 \pm 0.462$
$\mathbf{0.006 \pm 0.001}$
$\mathbf{7.050 \pm 0.436}$
$\mathbf{0.000 \pm 0.000}$ |
| SCOPE-Gen flipped | # Queries
Time
Set Size
Frac. Reject | –
–
–
– | $3.903 \pm 0.119$
$0.043 \pm 0.002$
$4.475 \pm 0.512$
$\mathbf{0.000 \pm 0.000}$ | $\mathbf{2.994 \pm 0.102}$
$0.041 \pm 0.002$
$2.326 \pm 0.140$
$\mathbf{0.000 \pm 0.000}$ | –
–
–
– |
| Non-Conformity: *count* | | | | | |
| CLM | # Queries
Time
Set Size
Frac. Reject | $6.562 \pm 0.197$
$2.300 \pm 0.024$
$3.855 \pm 1.104$
$\mathbf{0.000 \pm 0.000}$ | $20.000 \pm 0.000$
$84.599 \pm 9.140$
$18.646 \pm 1.637$
$0.003 \pm 0.000$ | $20.000 \pm 0.000$
$82.832 \pm 10.867$
$15.203 \pm 2.297$
$\mathbf{0.000 \pm 0.000}$ | $19.258 \pm 0.218$
$16.173 \pm 1.491$
$20.236 \pm 3.853$
$\mathbf{0.000 \pm 0.000}$ |
| CLM reduced max | # Queries | $4.204 \pm 0.114$ | $10.000 \pm 0.000$ | $10.000 \pm 0.000$ | $\mathbf{7.105 \pm 0.070}$ |

| Method | Metric | TriviaQA | MIMIC-CXR | CNN/DM | Molecules |
|---|---|---|---|---|---|
| | Time | $0.469 \pm 0.007$ | $9.922 \pm 0.088$ | $10.170 \pm 0.426$ | $0.405 \pm 0.008$ |
| | Set Size | $3.819 \pm 1.029$ | $18.257 \pm 1.709$ | $13.249 \pm 3.114$ | $17.753 \pm 0.000$ |
| | Frac. Reject | $\mathbf{0.000 \pm 0.000}$ | $0.163 \pm 0.000$ | $\mathbf{0.000 \pm 0.000}$ | $0.997 \pm 0.000$ |
| SCOPE-Gen config. 1 | # Queries | $\mathbf{2.754 \pm 0.204}$ | $\mathbf{3.974 \pm 0.063}$ | $\mathbf{3.172 \pm 0.186}$ | $7.896 \pm 0.485$ |
| | Time | $0.011 \pm 0.001$ | $0.040 \pm 0.002$ | $0.033 \pm 0.003$ | $0.011 \pm 0.001$ |
| | Set Size | $\mathbf{1.055 \pm 0.128}$ | $4.569 \pm 0.274$ | $2.538 \pm 0.451$ | $7.723 \pm 0.479$ |
| | Frac. Reject | $\mathbf{0.000 \pm 0.000}$ | $\mathbf{0.000 \pm 0.000}$ | $\mathbf{0.000 \pm 0.000}$ | $\mathbf{0.000 \pm 0.000}$ |
| SCOPE-Gen config. 2 | # Queries | $-$ | $5.039 \pm 0.449$ | $3.924 \pm 0.364$ | $-$ |
| | Time | $-$ | $0.011 \pm 0.003$ | $0.008 \pm 0.001$ | $-$ |
| | Set Size | $-$ | $4.288 \pm 0.703$ | $\mathbf{2.226 \pm 0.371}$ | $-$ |
| | Frac. Reject | $-$ | $\mathbf{0.000 \pm 0.000}$ | $\mathbf{0.000 \pm 0.000}$ | $-$ |
| SCOPE-Gen gen. only | # Queries | $3.633 \pm 0.036$ | $6.138 \pm 0.119$ | $5.525 \pm 0.297$ | $9.641 \pm 0.462$ |
| | Time | $\mathbf{0.005 \pm 0.000}$ | $0.004 \pm 0.000$ | $0.002 \pm 0.000$ | $\mathbf{0.005 \pm 0.000}$ |
| | Set Size | $1.722 \pm 0.191$ | $\mathbf{4.000 \pm 0.000}$ | $2.707 \pm 0.491$ | $\mathbf{7.074 \pm 0.486}$ |
| | Frac. Reject | $\mathbf{0.000 \pm 0.000}$ | $\mathbf{0.000 \pm 0.000}$ | $\mathbf{0.000 \pm 0.000}$ | $\mathbf{0.000 \pm 0.000}$ |
| SCOPE-Gen flipped | # Queries | $-$ | $4.750 \pm 0.530$ | $3.240 \pm 0.240$ | $-$ |
| | Time | $-$ | $0.028 \pm 0.004$ | $0.021 \pm 0.002$ | $-$ |
| | Set Size | $-$ | $5.815 \pm 1.368$ | $\mathbf{2.310 \pm 0.325}$ | $-$ |
| | Frac. Reject | $-$ | $\mathbf{0.000 \pm 0.000}$ | $\mathbf{0.000 \pm 0.000}$ | $-$ |
| | | | Non-Conformity: *max* | | |
| CLM | # Queries | $6.544 \pm 0.203$ | $20.000 \pm 0.000$ | $20.000 \pm 0.000$ | $19.267 \pm 0.212$ |
| | Time | $2.785 \pm 0.021$ | $111.025 \pm 15.329$ | $102.087 \pm 12.566$ | $15.301 \pm 1.092$ |
| | Set Size | $1.411 \pm 0.190$ | $10.699 \pm 1.959$ | $\mathbf{2.121 \pm 0.301}$ | $\mathbf{2.004 \pm 0.144}$ |
| | Frac. Reject | $\mathbf{0.000 \pm 0.000}$ | $\mathbf{0.000 \pm 0.000}$ | $\mathbf{0.000 \pm 0.000}$ | $\mathbf{0.000 \pm 0.000}$ |
| CLM reduced max | # Queries | $4.206 \pm 0.105$ | $10.000 \pm 0.000$ | $10.000 \pm 0.000$ | $\mathbf{7.107 \pm 0.066}$ |
| | Time | $0.985 \pm 0.010$ | $24.432 \pm 0.677$ | $26.455 \pm 1.557$ | $0.872 \pm 0.012$ |
| | Set Size | $1.499 \pm 0.304$ | $13.781 \pm 2.486$ | $2.401 \pm 0.426$ | $2.420 \pm 0.000$ |
| | Frac. Reject | $\mathbf{0.000 \pm 0.000}$ | $0.150 \pm 0.000$ | $0.003 \pm 0.000$ | $0.997 \pm 0.000$ |
| SCOPE-Gen config. 1 | # Queries | $\mathbf{3.056 \pm 0.098}$ | $\mathbf{3.922 \pm 0.305}$ | $\mathbf{3.379 \pm 0.156}$ | $7.891 \pm 0.532$ |
| | Time | $0.025 \pm 0.001$ | $0.061 \pm 0.008$ | $0.055 \pm 0.005$ | $0.077 \pm 0.003$ |
| | Set Size | $\mathbf{1.334 \pm 0.076}$ | $4.317 \pm 1.161$ | $2.697 \pm 0.144$ | $7.598 \pm 0.640$ |
| | Frac. Reject | $\mathbf{0.000 \pm 0.000}$ | $\mathbf{0.000 \pm 0.000}$ | $\mathbf{0.000 \pm 0.000}$ | $\mathbf{0.000 \pm 0.000}$ |
| SCOPE-Gen config. 2 | # Queries | $-$ | $5.096 \pm 0.744$ | $4.199 \pm 0.422$ | $-$ |
| | Time | $-$ | $0.021 \pm 0.007$ | $0.017 \pm 0.002$ | $-$ |
| | Set Size | $-$ | $\mathbf{5.761 \pm 0.980}$ | $3.078 \pm 0.575$ | $-$ |
| | Frac. Reject | $-$ | $0.160 \pm 0.000$ | $0.007 \pm 0.000$ | $-$ |
| SCOPE-Gen gen. only | # Queries | $3.633 \pm 0.036$ | $6.138 \pm 0.119$ | $5.525 \pm 0.297$ | $9.637 \pm 0.462$ |
| | Time | $\mathbf{0.006 \pm 0.000}$ | $\mathbf{0.005 \pm 0.000}$ | $\mathbf{0.002 \pm 0.000}$ | $\mathbf{0.006 \pm 0.001}$ |
| | Set Size | $3.024 \pm 0.080$ | $\mathbf{3.609 \pm 0.173}$ | $2.865 \pm 0.307$ | $7.025 \pm 0.418$ |
| | Frac. Reject | $\mathbf{0.000 \pm 0.000}$ | $\mathbf{0.000 \pm 0.000}$ | $\mathbf{0.000 \pm 0.000}$ | $\mathbf{0.000 \pm 0.000}$ |
| SCOPE-Gen flipped | # Queries | $-$ | $4.156 \pm 0.129$ | $3.384 \pm 0.123$ | $-$ |
| | Time | $-$ | $0.043 \pm 0.002$ | $0.040 \pm 0.002$ | $-$ |
| | Set Size | $-$ | $5.017 \pm 0.743$ | $3.315 \pm 0.368$ | $-$ |
| | Frac. Reject | $-$ | $\mathbf{0.000 \pm 0.000}$ | $\mathbf{0.000 \pm 0.000}$ | $-$ |
| | | | $\alpha = 0.4$ | | |
| | | | Non-Conformity: *sum* | | |
| CLM | # Queries | $6.555 \pm 0.196$ | $20.000 \pm 0.000$ | $20.000 \pm 0.000$ | $19.247 \pm 0.215$ |
| | Time | $2.724 \pm 0.018$ | $107.511 \pm 15.823$ | $100.752 \pm 12.215$ | $15.054 \pm 1.231$ |
| | Set Size | $1.397 \pm 0.258$ | $8.105 \pm 1.708$ | $2.816 \pm 0.500$ | $9.952 \pm 2.174$ |
| | Frac. Reject | $\mathbf{0.000 \pm 0.000}$ | $\mathbf{0.000 \pm 0.000}$ | $\mathbf{0.000 \pm 0.000}$ | $\mathbf{0.000 \pm 0.000}$ |
| CLM reduced max | # Queries | $4.207 \pm 0.103$ | $10.000 \pm 0.000$ | $10.000 \pm 0.000$ | $\mathbf{7.118 \pm 0.068}$ |
| | Time | $0.977 \pm 0.009$ | $23.955 \pm 0.596$ | $26.075 \pm 1.545$ | $0.864 \pm 0.011$ |
| | Set Size | $1.200 \pm 0.219$ | $10.039 \pm 2.115$ | $2.673 \pm 0.550$ | $9.639 \pm 1.911$ |
| | Frac. Reject | $\mathbf{0.000 \pm 0.000}$ | $0.003 \pm 0.000$ | $\mathbf{0.000 \pm 0.000}$ | $0.890 \pm 0.000$ |
| SCOPE-Gen config. 1 | # Queries | $\mathbf{2.674 \pm 0.147}$ | $\mathbf{3.724 \pm 0.343}$ | $\mathbf{2.698 \pm 0.033}$ | $7.600 \pm 0.443$ |
| | Time | $0.026 \pm 0.001$ | $0.052 \pm 0.004$ | $0.044 \pm 0.002$ | $0.096 \pm 0.004$ |
| | Set Size | $\mathbf{0.917 \pm 0.053}$ | $3.609 \pm 0.666$ | $\mathbf{1.379 \pm 0.058}$ | $5.864 \pm 0.219$ |
| | Frac. Reject | $\mathbf{0.000 \pm 0.000}$ | $\mathbf{0.000 \pm 0.000}$ | $\mathbf{0.000 \pm 0.000}$ | $\mathbf{0.000 \pm 0.000}$ |
| SCOPE-Gen config. 2 | # Queries | | $4.599 \pm 0.355$ | $3.653 \pm 0.316$ | |
| | Time | | $0.017 \pm 0.001$ | $0.015 \pm 0.001$ | |
| | Set Size | | $3.110 \pm 0.454$ | $1.578 \pm 0.277$ | |
| | Frac. Reject | | $\mathbf{0.000 \pm 0.000}$ | $\mathbf{0.000 \pm 0.000}$ | |
| SCOPE-Gen gen. only | # Queries | $3.633 \pm 0.036$ | $6.518 \pm 0.310$ | $5.525 \pm 0.297$ | $9.646 \pm 0.464$ |
| | Time | $\mathbf{0.006 \pm 0.000}$ | $\mathbf{0.002 \pm 0.000}$ | $\mathbf{0.002 \pm 0.000}$ | $\mathbf{0.006 \pm 0.001}$ |
| | Set Size | $1.484 \pm 0.021$ | $\mathbf{2.889 \pm 0.348}$ | $1.586 \pm 0.158$ | $\mathbf{5.540 \pm 0.168}$ |

| Method | Metric | TriviaQA | MIMIC-CXR | CNN/DM | Molecules |
|---|---|---|---|---|---|
| | Frac. Reject | **0.000 ± 0.000** | **0.000 ± 0.000** | **0.000 ± 0.000** | **0.000 ± 0.000** |
| SCOPE-Gen flipped | # Queries | – | 3.785 ± 0.156 | 2.775 ± 0.113 | – |
| | Time | – | 0.042 ± 0.002 | 0.040 ± 0.003 | – |
| | Set Size | – | 3.244 ± 0.468 | 1.486 ± 0.255 | – |
| | Frac. Reject | – | **0.000 ± 0.000** | **0.000 ± 0.000** | – |
| *Non-Conformity: count* | | | | | |
| CLM | # Queries | 6.552 ± 0.193 | 20.000 ± 0.000 | 20.000 ± 0.000 | 19.261 ± 0.223 |
| | Time | 2.301 ± 0.026 | 87.296 ± 15.006 | 83.194 ± 10.589 | 15.572 ± 1.655 |
| | Set Size | 4.690 ± 1.879 | 19.155 ± 1.166 | 12.349 ± 1.964 | 16.278 ± 4.197 |
| | Frac. Reject | **0.000 ± 0.000** | **0.000 ± 0.000** | **0.000 ± 0.000** | **0.000 ± 0.000** |
| CLM reduced `max` | # Queries | 4.200 ± 0.114 | 10.000 ± 0.000 | 10.000 ± 0.000 | **7.106 ± 0.065** |
| | Time | 0.471 ± 0.007 | 10.006 ± 0.127 | 10.306 ± 0.534 | 0.409 ± 0.008 |
| | Set Size | 4.790 ± 1.840 | 19.281 ± 1.183 | 11.934 ± 2.491 | 14.779 ± 3.371 |
| | Frac. Reject | **0.000 ± 0.000** | **0.000 ± 0.000** | **0.000 ± 0.000** | 0.833 ± 0.000 |
| SCOPE-Gen | # Queries | **2.661 ± 0.099** | **3.667 ± 0.076** | **2.937 ± 0.197** | 7.509 ± 0.474 |
| | Time | 0.010 ± 0.000 | 0.032 ± 0.003 | 0.026 ± 0.003 | 0.011 ± 0.001 |
| | Set Size | **0.938 ± 0.073** | 3.425 ± 0.449 | **1.708 ± 0.257** | 6.009 ± 0.450 |
| | Frac. Reject | **0.000 ± 0.000** | **0.000 ± 0.000** | **0.000 ± 0.000** | **0.000 ± 0.000** |
| SCOPE-Gen gen. only | # Queries | 3.633 ± 0.036 | 6.138 ± 0.119 | 5.525 ± 0.297 | 9.641 ± 0.462 |
| | Time | **0.005 ± 0.000** | **0.004 ± 0.000** | **0.002 ± 0.000** | **0.005 ± 0.001** |
| | Set Size | 1.553 ± 0.013 | 3.000 ± 0.000 | 2.007 ± 0.081 | **5.625 ± 0.272** |
| | Frac. Reject | **0.000 ± 0.000** | **0.000 ± 0.000** | **0.000 ± 0.000** | **0.000 ± 0.000** |
| SCOPE-Gen flipped | # Queries | – | 3.910 ± 0.061 | 3.091 ± 0.475 | – |
| | Time | – | 0.021 ± 0.001 | 0.019 ± 0.003 | – |
| | Set Size | – | 3.495 ± 0.120 | 1.853 ± 0.414 | – |
| | Frac. Reject | – | **0.000 ± 0.000** | **0.000 ± 0.000** | – |
| *Non-Conformity: max* | | | | | |
| CLM | # Queries | 6.574 ± 0.197 | 20.000 ± 0.000 | 20.000 ± 0.000 | 19.255 ± 0.210 |
| | Time | 2.816 ± 0.032 | 105.393 ± 12.837 | 101.326 ± 11.619 | 15.101 ± 0.937 |
| | Set Size | **1.170 ± 0.155** | 7.871 ± 1.612 | 1.772 ± 0.209 | **1.927 ± 0.097** |
| | Frac. Reject | **0.000 ± 0.000** | **0.000 ± 0.000** | **0.000 ± 0.000** | **0.000 ± 0.000** |
| CLM reduced `max` | # Queries | 4.207 ± 0.120 | 10.000 ± 0.000 | 10.000 ± 0.000 | **7.111 ± 0.066** |
| | Time | 0.987 ± 0.010 | 24.634 ± 0.634 | 26.286 ± 1.712 | 0.880 ± 0.013 |
| | Set Size | 1.176 ± 0.150 | 9.871 ± 2.349 | 1.838 ± 0.221 | 2.260 ± 0.219 |
| | Frac. Reject | **0.000 ± 0.000** | 0.003 ± 0.000 | **0.000 ± 0.000** | 0.857 ± 0.000 |
| SCOPE-Gen config. 1 | # Queries | **3.024 ± 0.103** | **3.711 ± 0.192** | 3.032 ± 0.089 | 7.515 ± 0.448 |
| | Time | 0.024 ± 0.001 | 0.052 ± 0.003 | 0.046 ± 0.002 | 0.076 ± 0.002 |
| | Set Size | 1.244 ± 0.098 | 3.622 ± 0.421 | 2.043 ± 0.078 | 5.942 ± 0.300 |
| | Frac. Reject | **0.000 ± 0.000** | **0.000 ± 0.000** | **0.000 ± 0.000** | **0.000 ± 0.000** |
| SCOPE-Gen config. 2 | # Queries | – | 4.732 ± 0.398 | 3.703 ± 0.293 | – |
| | Time | – | 0.008 ± 0.001 | 0.006 ± 0.001 | – |
| | Set Size | – | 3.279 ± 0.520 | **1.708 ± 0.263** | – |
| | Frac. Reject | – | **0.000 ± 0.000** | **0.000 ± 0.000** | – |
| SCOPE-Gen gen. only | # Queries | 3.633 ± 0.036 | 6.138 ± 0.119 | 5.525 ± 0.297 | 9.646 ± 0.464 |
| | Time | **0.005 ± 0.000** | **0.005 ± 0.001** | **0.002 ± 0.000** | **0.006 ± 0.002** |
| | Set Size | 2.935 ± 0.107 | **2.652 ± 0.158** | 2.247 ± 0.203 | 5.586 ± 0.229 |
| | Frac. Reject | **0.000 ± 0.000** | **0.000 ± 0.000** | **0.000 ± 0.000** | **0.000 ± 0.000** |
| SCOPE-Gen flipped | # Queries | – | 3.924 ± 0.375 | **3.019 ± 0.178** | – |
| | Time | – | 0.040 ± 0.002 | 0.037 ± 0.002 | – |
| | Set Size | – | 4.083 ± 0.791 | 1.855 ± 0.145 | – |
| | Frac. Reject | – | **0.000 ± 0.000** | **0.000 ± 0.000** | – |

# G  QUALITATIVE RESULTS

We show randomly chosen prediction sets from SCOPE-Gen (config. 1, see App. D.1) in comparison to prediction sets generated by CLM. We demonstrate examples for TriviaQA and MIMIC-CXR.

## G.1  TRIVIAQA

We calibrate both methods on a single (same) calibration set of size 1200 and $\alpha = 0.2$. Admissibility of an answer is denoted by the checkmark symbol ✓. If an answer is inadmissible, we denote it by ✗.

| Question | Prediction Set | |
|---|---|---|
| | **CLM** | **SCOPE-Gen** |
| What former U.S. president is known for his staunch support of Habitat for Humanity? | {Jimmy Carter (✓)} | {Jimmy Carter (✓)} |
| What cat food "tastes so good, cats ask for it by name"? | {Friskies (✗), Sheba (✗), Whiskas (✗), Fancy Feast (✗), Felix (✗), Purina (✗)} | {Friskies (✗), Whiskas (✗), Fancy Feast (✗), Sheba (✗), Felix (✗)} |
| What is the name of the giraffe that Toys-r-us uses as its' mascot? | {Geoffrey (✓), Geoffrey the Giraffe (✗), George (✗)} | {Geoffrey (✓)} |
| Where do you find the Bridal Veil, American, and Horseshoe Falls? | {Niagara Falls (✓), Niagara Falls, Canada (✓), Niagra Falls (✓), Niagara (✗)} | {Niagara Falls (✓), Niagara Falls, Canada (✓), Niagra Falls (✓)} |
| The worlds largest marketer of fruit juices, what is the juice arm of the Coca Cola company? | {Minute Maid (✓)} | {Minute Maid (✓)} |
| Whose backing band is known as The Miami Sound Machine? | {Gloria Estefan (✓), Gloria Estefan & Miami Sound Machine (✓), Gloria Estefan's (✗), Gloria Estafan (✓)} | {Gloria Estefan (✓), Gloria Estefan & Miami Sound Machine (✓)} |
| With a motto of Always Ready, Always There, what US military branch had it's founding on Dec 14, 1636? | {The Salvation Army (✗), US Marine Corps (✗), The Continental Army (✗), Marines (✗), The US Navy (✗), The National Guard (✓), The Coast Guard (✗), The Marine Corps (✗), The US Coast Guard (✗)} | {The National Guard (✓), The Coast Guard (✗), US Marine Corps (✗)} |
| Who tried to steal Christmas from the town of Whoville? | {The Grinch (✓)} | {The Grinch (✓)} |
| What is the name of the parson mentioned in the lyrics of the Christmas carol "Winter Wonderland"? | {Rev (✗), Father (✗), Frosty the Snowman (✗), Reverend Johnson (✗), Frosty (✗)} | {Father (✗), Rev (✗)} |
| In what outdoor sport, sanctioned by the NHPA, do you score 3 points for a ringer, 2 for a leaner, and the closet scores a point? | {Bocce (✗), Bowling (✗), Darts (✗), Ten-Pin Bowling (✗), Horseshoes (✓)} | {Bowling (✗), Darts (✗), Bocce (✗)} |

## G.2 MIMIC-CXR

We demonstrate predictions sets for a single instance (`study_id`: 59999516, `subject_id`: 13541358), where we calibrate SCOPE-Gen and CLM on a single (same) calibration set of size 1200 and $\alpha = 0.35$. Admissibility of an answer is denoted by the checkmark symbol ✓. We remove all newline characters in the generated answers for improved readability. We stress that outputs of both methods are sets and have no internal order. We index answers for the purpose of obtaining identifiers for the different predictions.

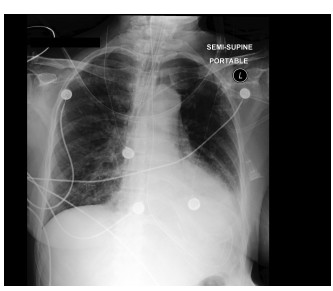

| | **Prompt** | |
|---|---|---|
| | FINAL REPORT
EXAMINATION: CHEST (PORTABLE AP)

INDICATION: ___ year old woman with AMS of unclear cause, HCAP // Evaluate
interval change in RLL infiltrate Evaluate interval change in RLL infiltrate

IMPRESSION: | |

| | **Reference Report ("Ground Truth")** | |
|---|---|---|
| | Compared to prior chest radiographs ___ through ___.

New consolidation in the left lower lobe is accompanied by ipsilateral mediastinal shift. Although that indicates a component of atelectasis, there could be concurrent pneumonia in the left lower lobe particularly if patient has aspirated. Right lung is clear. Heart size top-normal. Left pleural effusion is presumed, but not large.

Left PIC line ends in the mid SVC. ET tube tip is one cm below optimal placement with the chin down. Esophageal drainage tube ends in a nondistended stomach. | |

| | **SCOPE-Gen** | |
|---|---|---|
| **Index** | **Generated** | **Admissible** |
| 1 | Comparison to ___. The endotracheal tube has been slightly advanced. The tip of the device now projects approximately 5 cm above the carina. Mild retrocardiac atelectasis persists. Mild fluid overload but no overt pulmonary edema. No larger pleural effusions. Borderline size of the cardiac silhouette. The nasogastric tube has been removed. | ✓ |

*Continued on next page*

| Index | Generated | Admissible |
|---|---|---|
| 2 | In comparison with the study of ___, the monitoring and support devices are essentially unchanged. Continued enlargement of the cardiac silhouette with pulmonary vascular congestion. Opacification at the left base is consistent with volume loss in the left lower lobe and probable small left pleural effusion. | ✓ |
| 3 | Compared to chest radiographs ___ through ___. Left lower lobe is still collapsed, either pneumonia or atelectasis. This could be the result of aspiration on ___ and subsequent development of lung abscess. Pleural effusion is minimal, if any. No pneumothorax. Heart is top-normal size. ET tube in standard placement. Nasogastric tube ends in the upper stomach. Left subclavian line ends in the upper right atrium. | ✗ |
| 4 | Tracheostomy is in place. Heart size and mediastinum are stable. Left lower lobe consolidation is present. Substantial left upper lobe atelectasis and more pronounced in the left perihilar region are unchanged. There is no pneumothorax. | ✓ |
| 5 | Comparison to ___. The lung volumes have increased, the areas of atelectasis, restricted to the left and right upper lobe continues to be minimally improved. Decrease in extent of a retrocardiac atelectasis. Signs of mild fluid overload but no overt pulmonary edema. . Stable size of the cardiac silhouette. | ✗ |

CLM

| Index | Report | Admissible |
|---|---|---|
| 1, 2, 3, 4, 5 | identical to 1, 2, 3, 4, 5 from SCOPE-Gen | see above |
| 6 | Compared to chest radiographs ___ through ___. ET tube, left PIC line, and upper enteric drainage tube all in standard placements. Left lower lobe atelectasis unchanged, responsible for large leftward mediastinal shift. Mild cardiomegaly unchanged. Pulmonary vascular engorgement slightly improved since ___. Pleural effusions small if any. No evidence of pneumothorax. | ✓ |
| 7 | In comparison with the study of ___, there is little overall change except for increase in opacification at the left base consistent with some combination of pleural fluid and volume loss in the left lower lobe with underlying consolidations. The upper zone vessels are more prominent, consistent with greater vascular congestion. | ✓ |
| 8 | ET tube tip is 2 cm above the carinal. Left PICC line tip is at the level of mid SVC. NG tube tip passes below the diaphragm and most likely is in the stomach. Heart size and mediastinum are unchanged compared to previous study. The left upper lobe is collapsed. Small left pleural effusion is most likely present. | ✓ |
| 9 | Compared to chest radiographs ___ through ___. Previous mild to moderate pulmonary edema improved and left lower lobe atelectasis has decreased. Heart size top-normal unchanged. Pleural effusion small if any. No pneumothorax. ET tube, left PIC line, and esophageal drainage tubes are in standard positions. | ✓ |
| 10 | ET tube tip is 3.3 cm above the carinal. NG tube tip is in the stomach. Left PICC line tip is at the level of the mid to lower SVC. Hiatal hernia is still present. Since the prior study pulmonary vascular congestion has improved. Small amount of bilateral pleural effusion is more pronounced on the left than on the right, consistent with improved pulmonary vascular congestion. No pneumothorax is seen. | ✓ |

| Index | Generated | Admissible |
|---|---|---|
| 11 | In comparison with the study of ___, there is further increase in the opacification at the right base with silhouetting hemidiaphragm, consistent with worsening pneumonia or effusion or even pulmonary edema. Endotracheal tube has been removed. The left subclavian catheter again extends to about the level of the carina. | ✗ |
| 12 | ET tube tip is approximately 5 cm above the carinal pointing toward the right apex. NG tube tip is in the stomach. There is no pneumothorax. NOTIFICATION: The findings were discussed with ___, M.D. by ___ ___, M.D. on the telephone on ___ at 10:07 AM, 2 minutes after discovery of the findings. | ✗ |
| 13 | Comparison to ___. The lung volumes have increased, the size of the cardiac silhouette is now at the upper range of normal, there has been development of a left basilar opacity. These abnormalities are visible with elevation of the left hemidiaphragm and the presence of small left pleural effusion. The right lung is moderately well inflated. No pleural effusions. No pneumonia. | ✗ |
| 14 | ET tube tip is 2.5 cm above the carinal. NG tube tip is in the stomach. Heart size is top-normal. Mediastinum is overall unremarkable with slight deviation of the left minor fissure to the left. There is no pneumothorax. | ✗ |
| 15 | No previous images. Endotracheal tube tip lies approximately 5 cm above the carina. Cardiomediastinal silhouette is within normal limits. Left pleural effusion with compressive basilar atelectasis, worse on the left, is essentially unchanged. | ✓ |
| 16 | Comparison to ___. There is minimally improved ventilation in the retrocardiac lung region. Otherwise the radiograph is unchanged. The patient has received an endotracheal tube and a nasogastric tube. | ✗ |
| 17 | ET tube tip is 3.5 cm above the carina. NG tube tip is in the stomach. There is left retrocardiac consolidation that could represent consolidation in the left lower lobe or infrahilar consolidation due to pneumonia. Followup radiograph is recommended. RECOMMENDATION(S): Followup radiograph is recommended. | ✗ |
| 18 | Compared to chest radiographs ___ through ___ at 07:47. Large left lower lobe pneumonia unchanged over several days. Pulmonary vascular congestion now appears mild. Heart size normal. No pneumothorax. ET tube in standard placement. Nasogastric drainage tube would need to be advanced at least 15 cm to move all the side ports into the stomach, if any. | ✗ |
| 19 | As compared to ___, no relevant change is seen. Improved ventilation of the left lower lobe. Unchanged size of the cardiac silhouette. | ✗ |

## H    POST-EVALUATING ADMISSIBILITY

If a human domain expert is used to assess admissibility, we recommend post-evaluating the achieved admissibility in the following way: First, set a certain amount from the calibration data $\{x_i\}_{i=1}^n$ aside, such that we have two data sets

$$\{x_i\}_{i=1}^m \quad \text{and} \quad \{x_i\}_{i=m+1}^n. \tag{21}$$

Second, calibrate SCOPE-Gen on $\{x_i\}_{i=1}^m$. Finally, use the test set to generate prediction sets, using the calibrated parameters. Query the admissibility function to assess the fraction of admissible

prediction sets:

$$\hat{A} := \sum_{i=m+1}^{n} \frac{\mathbb{1}\{A(\mathcal{C}_{\boldsymbol{\lambda}^*}(x_i)) = 1\}}{n - m}. \tag{22}$$

This fraction $\hat{A}$ provides an unbiased estimate of the achieved admissibility, conditionally on the calibration data.

Choosing three different values for $m$, *sum* non-conformity and three different levels for $\alpha$ and $n$ always such that $n - m = 300$, we can get the following admissibility estimates (that are conditional on the calibration data):

| **TriviaQA** | | | |
|---|---|---|---|
| | $\alpha = 0.3$ | $\alpha = 0.35$ | $\alpha = 0.4$ |
| $m = 300$ | $0.86 \pm 0.35$ | $0.69 \pm 0.46$ | $0.65 \pm 0.48$ |
| $m = 600$ | $0.77 \pm 0.42$ | $0.71 \pm 0.46$ | $0.67 \pm 0.47$ |
| $m = 1200$ | $0.69 \pm 0.46$ | $0.63 \pm 0.48$ | $0.58 \pm 0.49$ |

| **MIMIC-CXR** | | | |
|---|---|---|---|
| | $\alpha = 0.3$ | $\alpha = 0.35$ | $\alpha = 0.4$ |
| $m = 300$ | $0.71 \pm 0.45$ | $0.66 \pm 0.47$ | $0.61 \pm 0.49$ |
| $m = 600$ | $0.67 \pm 0.47$ | $0.62 \pm 0.49$ | $0.61 \pm 0.49$ |
| $m = 1200$ | $0.71 \pm 0.45$ | $0.65 \pm 0.48$ | $0.57 \pm 0.50$ |

| **CNN-DM** | | | |
|---|---|---|---|
| | $\alpha = 0.3$ | $\alpha = 0.35$ | $\alpha = 0.4$ |
| $m = 300$ | $0.70 \pm 0.46$ | $0.67 \pm 0.47$ | $0.55 \pm 0.50$ |
| $m = 600$ | $0.70 \pm 0.46$ | $0.65 \pm 0.48$ | $0.58 \pm 0.49$ |
| $m = 1200$ | $0.73 \pm 0.45$ | $0.68 \pm 0.47$ | $0.61 \pm 0.49$ |

| **Molecules** | | | |
|---|---|---|---|
| | $\alpha = 0.3$ | $\alpha = 0.35$ | $\alpha = 0.4$ |
| $m = 300$ | $0.75 \pm 0.43$ | $0.71 \pm 0.46$ | $0.67 \pm 0.47$ |
| $m = 600$ | $0.73 \pm 0.44$ | $0.66 \pm 0.47$ | $0.63 \pm 0.48$ |
| $m = 1200$ | $0.70 \pm 0.46$ | $0.65 \pm 0.48$ | $0.62 \pm 0.49$ |

In this manner, we can empirically assess the achieved admissibility post calibration.

# I DISCUSSION & LIMITATIONS

**Data Splitting.** SCOPE-Gen partitions calibration data across the prediction steps to control the factors of the Markov chain (7). However, this segmentation leads to increased variance and looser conformal prediction sets due to reduced calibration set sizes at each step. A promising alternative could involve controlling the total risk through family-wise error rate control (Benjamini & Braun, 2002). This approach could be particularly advantageous when combined with our sequential filtering strategy, which only requires accounting for $K$ distinct tests rather than the $g^K$ tests required by the method of Quach et al. (2024) (where $K$ is the amount of calibration parameters and $g$ is the amount of points on a parameter grid required for CLM).

**Conformal Prediction vs. Learn-then-Test.** In general, the learn-then-test framework is more flexible than conformal prediction as it does not require assumptions of monotonicity or uni-dimensional calibration parameters. Nonetheless, the present work demonstrates that reformulating the given problem into multiple conformal prediction problems has benefits (as shown experimentally § 5). In addition, the *intuitive* admissibility guarantee of conformal prediction (1) requires specification of a single parameter only, in comparison to the *complex* two-parameter guarantee of learn-then-test (12). While the latter guarantee bounds the conformal admissibility guarantee, we note that the optimization strategy employed in our experiments (see § 5) would in practice have to be conducted on a separate data set (due to data contamination), leading to decreased sample efficiency.

**Generality vs. Computational Demand.** SCOPE-Gen is applicable for a wide variety of generative models, because it assumes access to i.i.d. samples, which are available for virtually any type of generative model. During inference, however, SCOPE-Gen presents a computational overhead in comparison to i.i.d. sampling. This fact can be traced back to the sequential filters, which remove some of the initial i.i.d. predictions. In settings where computational demand during inference is a concern, we note that our method could be combined with architecture-specific approaches for ensuring diversity (e.g., Corso et al. (2023); Kirchhof et al. (2024); Vilnis et al. (2023)) and high quality samples (e.g., by sampling at low temperature), in order to reduce the amount of examples that are filtered out. Assuming that the sampling is performed consistently during calibration and for the test point ($X_{n+1}$, see (1)), such sampling techniques do not affect the admissibility control guarantee (1).

