# OpenReview forum: "Conformal Generative Modeling with Improved Sample Efficiency through Sequential Greedy Filtering"
_ICLR.cc/2025/Conference — ICLR 2025 Poster_

### Official Review · Reviewer_7ufb · 2024-11-04

**Soundness:** 2
**Presentation:** 3
**Contribution:** 2
**Rating:** 6
**Confidence:** 1

**Summary:**

This manuscript introduces a sequential conformal prediction method called SCOPE-Gen. SCOPE-Gen uses a sequential pruning approach to iteratively refine the prediction set, allowing for separate control over each factor in the Markov chain, and demonstrates a significant reduction in the number of admissibility evaluations required during calibration. The method has been experimentally validated in natural language generation and molecular graph extension tasks.

**Strengths:**

Generally well-written; it’s clear that the manuscript is largely inspired (and adopted) from the setup in Quach et al 2024. Nevertheless, the authors detailed differences to Quach et al 2024 and highlight the efficiency of their method by leveraging sequential factorization.

**Weaknesses:**

- (minor) Fig 1 is slightly unclear — the caption should at least include some explanation of \nu, which is not specified until Sec 3

**Questions:**

- Can the author clarify the results in Table 2 of the appendix? It appears that the performance gap between SCOPE and CLM is narrowing - can the authors explain why this might be happening?

---

> ### Author Response · Authors · 2024-11-19
>
> We appreciate the time and effort taken to review our manuscript. We are happy to incorporate the reviewer's remarks into the revised version of our manuscript.
>
> We address the listed weaknesses and questions one-by-one:
>
> > (minor) Fig 1 is slightly unclear — the caption should at least include some explanation of \nu, which is not specified until Sec 3
>
> We thank the reviewer for catching this detail. We have updated the caption of Figure 1 in the revised manuscript. It is now explicitly specified what $\nu$ is.
>
> > Can the author clarify the results in Table 2 of the appendix? It appears that the performance gap between SCOPE and CLM is narrowing - can the authors explain why this might be happening?
>
> We are unsure about what the reviewer means by "narrowing gap" between SCOPE and CLM in Table 2. Table 2 highlights that SCOPE consistently requires fewer admissibility checks compared to CLM across all our tasks. It would be great if the reviewer could provide additional details/clarification on this remark.
>
> We thank the reviewer once again for supporting us in improving our manuscript.

---

### Official Review · Reviewer_fuTF · 2024-11-05

**Soundness:** 2
**Presentation:** 3
**Contribution:** 2
**Rating:** 6
**Confidence:** 2

**Summary:**

The paper proposes a heuristic algorithm to filter predictions of generative model to achieve conformal admissability. It argues that previous techniques need to evaluate the admissability multiple times per instance during the calibration phase, which is impractical when the admissability is evaluated by a human oracle. In their setup, the admissability factorizes into Markov chain and thus es requires fewer queries to the admission function (e.g. human oracle). The paper presents the algorithm for the filtering heuristic as well as the necessary calibration algorithms using two filters based on diversity and quality functions of the generated examples. The experiments over natural language generative tasks and molecular generation reporting favorable metrics in particular in terms of reduction in number of queries and runtime.

**Strengths:**

The paper addresses an important problem of current stochastic generative models hallucinating non-factual responses. Formulating this problem within the risk-control framework can provide the mathematical means for addressing it. The experimental evaluation over the natural langauge tasks seems relevant.

**Weaknesses:**

I find the empirical evaluation very difficult to asses both, on its own as well as in comparison to previous methods. To understand the effects of various parts of the proposed algorithms, it would be beneficial to perform ablation studies that could provide more insights into the effects of its individual components (e.g. the update function, the coverage level $\alpha$, etc.).
I am not convinced about the benefits of the molecular example - when there is only one valid/admissable example, it seems to me that simple checking of the validity at the generation time for each generated specimen should be enough. I do not see the benefit of the proposed method in this setup. This also applies to the TriviaQA experiment.
The algorithm requires an independent calibration set which seems to be very difficult to obtain in practice. In the presented experiments either boils down to something very trivial (single example being the valid one) or relying on another model which itself may be of uncertain quality. Further, I see similar issue with the update and filter functions which seem difficult to formulate reasonable in realistic scenarios. For me these are major limitations of the method which shall be discussed.

The main text of the paper (section 9) spills over to page 11. As per the call for papers, there is a strict 10 page limit on the main text and the call suggests a desk reject in case of violations of this limit.

**Questions:**

1. One of the motivations for the method is the possible reliance on human oracle and expenses related to querying it. In the end the experiment use a non-human validation function. Related questions:
- Would not the human oracle make some of the assumptions invalid (e.g. the need for increasing update function)?
- As some point you mention that multiple queries over the same example need to be executed - wouldn't the human validation bring even more noise into the whole process and invalidate some of your probabilistic conclusions?
2. I do not understand equation (6). What level of quantile is this? What is the interpretation of the n-fracion in the right-hand side?
3. Is there a way to independently post-evaluate that the experimental results are really conformal with the $\alpha$ level you were trying to achieve. Or would you need to use your own calibration parameters? If it were possible, this would provide additional useful insight.
4. Please address the concerns mentioned under Weaknesses.

---

> ### Author Response · Authors · 2024-11-19
>
> We appreciate the time and effort taken to review our manuscript. The reviewer's feedback has highlighted areas where we can improve our explanations, and we are grateful for the opportunity to address all concerns.
>
> Below, we address the listed weaknesses and questions one-by-one:
>
> > I find the empirical evaluation very difficult to asses both, on its own as well as in comparison to previous methods. To understand the effects of various parts of the proposed algorithms, it would be beneficial to perform ablation studies that could provide more insights into the effects of its individual components (e.g. the update function, the coverage level , etc.)
>
> We agree. To provide insights into the effects of the various components, we have included comprehensive ablation studies in Appendix F (of the revised manuscript), analyzing different update functions, filter usage, and filter ordering. In the revised manuscript, we ensure these studies are more prominently referenced in the main text (specifically, in Section 5) to aid in assessment. Additionally, we have conducted new experiments with varying admissibility levels $\alpha \in \\{ 0.3, 0.35, 0.4 \\}$, which we have included in the revised manuscript.
>
> > I am not convinced about the benefits of the molecular example - when there is only one valid/admissable example, it seems to me that simple checking of the validity at the generation time for each generated specimen should be enough. I do not see the benefit of the proposed method in this setup. This also applies to the TriviaQA experiment.
>
> To clarify, admissibility of the generated examples *cannot be assessed at generation time* for these tasks (and all other tasks). We can make this more concrete using TriviaQA: In the calibration set, the textual question correspecificallysponds to $x$ and the correct answer corresponds to $y^t$. The calibration set is a set of pairs $\\{ (x_i, y^t_i) \\}\_{i=1}^n $. This means that we can check for admissibility in the calibration set. At test time, however, admissibility *cannot* be checked. At test time, we only have access to the question $x_{n+1}$, but *not* the correct answer $y^t_{n+1}$. Thus, answers cannot be checked for admissibility. We have added a footnote on page 3 to clarify this point.
>
> > The algorithm requires an independent calibration set which seems to be very difficult to obtain in practice
>
> In many real-world scenarios, calibration sets can be effectively obtained: For data sets where admissibility corresponds to an exact match, the calibration set can simply be obtained by setting some data aside from the training set. If admissibility does not correspond to an exact match (such as for a radiology report), admissibility should ideally be evaluated by a human domain-expert. While this is indeed costly, our method makes a great improvement over prior work, as clearly demonstrated in our experiments (see subsection 5.2).
>
> > In the presented experiments either boils down to something very trivial (single example being the valid one) or relying on another model which itself may be of uncertain quality.
>
> As mentioned earlier, admissibility cannot be assessed at generation time, making our method essential even when only one valid example exists. In theory, this admissibility function can be any function, and the guarantee will hold for that specific function (as given by a machine or human). We use automated admissibility checks for experimental purposes only (see first paragraph of section 5.1). In practice, we believe that admissibility must be assessed by a human domain expert to prevent such uncertainty in the quality.
>
> > Further, I see similar issue with the update and filter functions which seem difficult to formulate reasonable in realistic scenarios
>
> We demonstrate our method in the context of radiology report generation (Section 6, experiment 2), which is a highly relevant and realistic application/scenario. The same functions can be applied in different domains such as summarizing news articles (experiment 3), without requiring reformulations. We hope that this explanations clarifies concerns about the practicality in realistic scenarios and are happy to get back to this point if not.
>
> > The main text of the paper (section 9) spills over to page 11. As per the call for papers, there is a strict 10 page limit on the main text and the call suggests a desk reject in case of violations of this limit.
>
> We assure the reviewer that the main text ends on page 10. The content on page 11 is the reproducibility statement, which complies with the conference guidelines and does not count towards the page limit. We cite the [ICLR 2025 Author Guidelines](https://iclr.cc/Conferences/2025/AuthorGuide):
>
> > This optional reproducibility statement will not count toward the page limit, but should not be more than 1 page.

---

> > ### Author Response · Authors · 2024-11-19
> >
> > We furthermore address all questions:
> >
> > > 1. One of the motivations for the method is the possible reliance on human oracle and expenses related to querying it. In the end the experiment use a non-human validation function. Related questions: Would not the human oracle make some of the assumptions invalid (e.g. the need for increasing update function)?
> >
> > The use of a human oracle does not invalidate our assumptions because our framework does not impose restrictions on the admissibility function, whether it is automated or human-based (see Section 2). The need for an increasing update function remains applicable, and the theoretical guarantees of our method hold under these conditions. To clarify this point, we have added explanations on pages 3 and 8 in the revised manuscript.
> >
> > > As some point you mention that multiple queries over the same example need to be executed - wouldn't the human validation bring even more noise into the whole process and invalidate some of your probabilistic conclusions?
> >
> > In our framework, we assume that human domain experts provide consistent evaluations of admissibility, especially in specialized fields like medicine where professionals are specifically trained to make accurate judgments.
> >
> > > 2. I do not understand equation (6). What level of quantile is this? What is the interpretation of the n-fracion in the right-hand side?
> >
> > We agree that this equation may not be clear for readers who are not familiar with conformal prediction. We therefore decided to include an additional section to our appendix (Appendix B) where we provide a formal definition of the empirical quantile. In addition, we added a footnote to clarify the background for the $\lceil (1 - \alpha)(n + 1) \rceil / n$ term.
> >
> > > 3. Is there a way to independently post-evaluate that the experimental results are really conformal with the level you were trying to achieve. Or would you need to use your own calibration parameters? If it were possible, this would provide additional useful insight.
> >
> > We thank the reviewer for this insightful question. If a human domain expert is used to assess admissibility, we recommend post-evaluating the achieved admissibility in the following way: First, set a certain amount from the calibration data aside (let us refer to it as "test set"). Then, calibrate SCOPE-Gen on the rest of the calibration data. Finally, use the test set to generate prediction sets, using the calibrated parameters. Query the admissibility function to assess the fraction of admissible prediction sets. This fraction provides an unbiased estimate of the achieved admissibility, conditionally on the calibration data.
> >
> > We decided to integrate this recommended procedure into the Appendix (Appendix H of the revised manuscript). We also demonstrate the results of this procedure for our data sets and method.
> >
> > > 4. Please address the concerns mentioned under Weaknesses.
> >
> > We hope that our responses have satisfactorily addressed the concerns. We are committed to improving our manuscript and appreciate the thoughtful feedback.

---

> ### Comment · Reviewer_fuTF · 2024-11-24
>
> Dear authors, thank you for your responses. Some further thoughts.
>
> * Ablations - thank you for these. Yes, these indeed provide some additional useful insights. The Scope-Gen gen only seems to perform very well and I believe it would be worth to provide the reader with more info to help understanding, what's happenning. Why do you explore the addmissibility level in such a narrow interval (0.3-0.4)? Would  significantly lower or higher values behave similarly? I may have missed it but is there any conclusion / recommendation for the count/sum/max measures?
> * Admissibility - ah, I see, ok
> * How to obtain calibration sets - would be worth mentioning in the main text
> * Trivial problems - ah, I see, ok
> * Update and filter funcitons - ok
> * Human evaluation - ok
> * quantile - ok
> * post-evaluation - good.
>
> I appreciate the clarificaitons and I find that the updates improved the paper. I have increased my score to weak accept.

---

> ### Author Response · Authors · 2024-11-26
>
> We are happy to hear that the reviewer appreciates our clarifications. We would like to add three more remarks regarding the response:
>
> > Yes, these indeed provide some additional useful insights. The Scope-Gen gen only seems to perform very well and I believe it would be worth to provide the reader with more info to help understanding, what's happenning.
>
> We have included a brief dicussion about the additional experiments in Appendix F.2. There exists another SCOPE-Gen configuration that works better in terms of minimizing prediction set size for MIMIC-CXR and CNN/DM, at the cost of slightly more required admissibility checks. We have decided to include this "configuration 2" in the experiments of our revised manuscript.
>
> > Why do you explore the addmissibility level in such a narrow interval (0.3-0.4)? Would significantly lower or higher values behave similarly?
>
> Much lower admissibility levels would not behave similarly, because the performance of the underlying model sets a limit on which admissibility levels are theoretically possible to achieve. For example, if a model does not achieve a single admissible answer within the $\texttt{max}$ amount of tries for more than $30 \\%$ of questions, we see that admissible sets cannot be achieved for level $\alpha < 0.3$, independent of the method that is used. Thus, setting $\alpha$ even lower will result in rejected calibrations, as outlined in Section 4.2. We thus decided to only demonstrate a range for $\alpha$ that is theoretically achievable across all experiments. We decided to elucidate on this point in Appendix F.2 of our revised manuscript. We thank the reviewer for bringing this to our attention.
>
> > I may have missed it but is there any conclusion / recommendation for the count/sum/max measures?
>
> We thank the reviewer for this question. In general, the optimal non-conformity measure depends on the experiment and admissibility level. It is thus difficult to make a general recommendation. We decided to include a small discussion around this point in Appendix F.2, using the experimental results.

---

### Official Review · Reviewer_WrdN · 2024-11-06

**Soundness:** 3
**Presentation:** 3
**Contribution:** 2
**Rating:** 5
**Confidence:** 3

**Summary:**

The paper presents SCOPE-Gen, a sequential conformal prediction method designed to generate prediction sets that satisfy admissibility guarantees with high probability. The method operates in two main stages: a generation stage and a filtering stage. In the generation stage, i.i.d. samples are drawn from the generative model until a specified non-conformity measure (related to sample count or quality) surpasses a threshold set by calibration with ground-truth samples. In the filtering stage, the prediction set is refined in a greedy manner, optimizing for diversity and quality based on another threshold derived from calibration. To ensure admissibility, the approach leverages a Markov chain factorization for admissibility control, and calibration is conducted on independent, non-overlapping data subsets to enable this factorization. Experimental results demonstrate that SCOPE-Gen reduces both the number of queries to the admission function during calibration and the size of the prediction set needed to meet admissibility requirements, outperforming baseline methods like CLM.

**Strengths:**

* This paper presents an efficient approach for generating prediction sets with admissibility guarantees by using a sequential generation and greedy filtering strategy.
* It reduces the number of admissibility checks during calibration compared to previous baselines, improving computational efficiency.
* Experimental results support the method’s effectiveness in reducing both query counts and prediction set sizes to meet admissibility criteria.

**Weaknesses:**

* The paper lacks a theoretical analysis detailing how effectively the proposed method reduces the required admissibility checks and prediction set size.
* The sequential generation and filtering process may introduce additional computational costs by generating a large number of samples before the filtering stage.
* The calibration process, which involves sample splitting for generation and each filtering stage, may require extra ground-truth samples to determine accurate threshold (lambda) values.

**Questions:**

* The paper defines admissibility as including at least one (semantically) correct answer in the prediction set and aims to minimize the prediction set size while ensuring this inclusion. This is achieved by a “sub-sampling” technique, sampling answers based on a quality score ranking. Can the proposed method generalize to a broader admissibility definition, such as including multiple correct answers (e.g., 5 out of 10) or maximizing the fraction of correct answers? How would this method perform compared to baselines if the goal were to optimize the fraction of acceptable answers within a fixed prediction set size?
* How does the sequence of filtering stages (diversity vs. quality) impact performance? Why is diversity filtering prioritized in the proposed method?

---

> ### Author Response · Authors · 2024-11-19
>
> We appreciate the time and effort taken to review our manuscript. We are glad to incorporate the comments and suggestions into the revised version.
>
> We address the listed weaknesses and questions one-by-one:
>
> > The paper lacks a theoretical analysis detailing how effectively the proposed method reduces the required admissibility checks and prediction set size.
>
> We note that the amount of required admissibility checks is upper bounded by the amount of checks required for CLM. Regarding theoretical analysis, we observe that the mentioned effectiveness depends highly on the used non-conformity measure. Thus, a theoritical analysis is likely to be bounded to a very specific setting without possibilities for generalization.
>
> > The sequential generation and filtering process may introduce additional computational costs by generating a large number of samples before the filtering stage.
>
> We appreciate the reviewer's correct observation. Although the filtering process does introduce a computational overhead over standard i.i.d. sampling, we believe it is justified by the significant advantages it offers. The filtering step enables our method to work with any generative model (generating i.i.d. samples) without requiring modifications to the model architecture, thus providing broad applicability.
>
> If avoiding computational cost is of great importance, one potential strategy would be to adjust the sampling parameters of the generative model. For instance, sampling from a language model at a lower temperature can reduce the generation of low-quality examples. Additionally, existing methods for generating diverse (non-i.i.d.) prediction sets (e.g., [1, 2]) could be integrated with our approach. In such cases, the filtering steps will remove fewer examples, while still maintaining the admissibility control guarantee.
>
> We recognize the trade-off between computational cost and generality of our method. To address this point, we have incorporated a discussion paragraph called "Computational Demand vs. Generality" (Appendix I) of the revised manuscript. We thank the reviewer for bringing this important point to our attention.
>
> > The calibration process, which involves sample splitting for generation and each filtering stage, may require extra ground-truth samples to determine accurate threshold (lambda) values.
>
> This is a good point and it is discussed in Section 7 (now Appendix I in the revised manuscript). However, we would like to stress that our method tends to yield better empirical results than prior work, *in spite of data splitting*. Considering the vast amount of data required to train/fine-tune a generative model, we believe that setting aside a data sample of size around $n=600$ should not be of significant practical concern.
>
> [1] Corso, Gabriele, et al. "Particle Guidance: Non-IID Diverse Sampling with Diffusion Models." International Conference on Learning Representations (2023).
>
> [2] Vilnis, Luke, et al. "Arithmetic Sampling: Parallel Diverse Decoding for Large Language Models." International Conference on Machine Learning (2023).

---

> > ### Author Response · Authors · 2024-11-19
> >
> > We would furthermore like to address the reviewer's questions:
> >
> > > The paper defines admissibility as including at least one (semantically) correct answer in the prediction set and aims to minimize the prediction set size while ensuring this inclusion. This is achieved by a “sub-sampling” technique, sampling answers based on a quality score ranking. Can the proposed method generalize to a broader admissibility definition, such as including multiple correct answers (e.g., 5 out of 10) or maximizing the fraction of correct answers? How would this method perform compared to baselines if the goal were to optimize the fraction of acceptable answers within a fixed prediction set size?
> >
> > One general objective of conformal prediction methods (including our method) is to report small prediction sets, as correctly pointed out by the reviewer. Our method is not designed for obtaining guarantees in terms of a fraction of correct predictions in the prediction set. We furthermore note that such a guarantee is theoretically impossible to provide, in general. To see this, we imagine a generative model that never makes a correct prediction. What SCOPE-Gen will do in this case is to reject the calibration and return the entire output space $\mathcal{Y}$ such that admissibility is (trivially) guaranteed. However, this cannot be done for a guarantee that corresponds to returning at least a fraction of correct answers, because there is no prediction set that trivially satisfies the proposed constraint.
> >
> > In addition, we are uncertain about the practical benefit of the proposed guarantee because a method that ensures the guarantee would certainly not be able to inform the user about which ones of the generated outputs belong to the "correct fraction" of the prediction set (if that would be possible, we would not need the method to begin with). In contrast, our method provides a valuable guarantee in the sense of a worst-case scenario by ensuring that at least one of the predictions in the prediction set is correct, with high probability. This is particularly useful in applications where missing a correct answer could have significant consequences. Users can proceed with counteracting the prediction(s) that would entail the most severe consequences, even if they cannot identify the admissible answer(s) in the prediction set.
> >
> > > How does the sequence of filtering stages (diversity vs. quality) impact performance? Why is diversity filtering prioritized in the proposed method?
> >
> > We thank the reviewer for pointing out the relevance of ablation studies. We have conducted an ablation study where we flip the order of the filters and an ablation study where we ommit filtering alltogether (thus, only performing the generation step). The results of these experiments can be inspected in Appendix F.2. We have added a remark to the main text to highlight more explicitly that such ablation studies are indeed performed (see first paragraph of Section 5 in the revised manuscript). We furthermore apologize if our description gave the impression that the diversity filter is prioritized. We would be happy to provide further clarification on this aspect if needed.
> >
> > We are committed to improving our manuscript and thank the reviewer once again for the thoughtful feedback.

---

### Official Review · Reviewer_XFHD · 2024-11-10

**Soundness:** 3
**Presentation:** 3
**Contribution:** 3
**Rating:** 8
**Confidence:** 4

**Summary:**

This fairly well-explicated paper considers the following question of much recent interest: how could we obtain some semblance of guarantees for generative models' outputs i.e. in terms of factuality. This is related to the problem of hallucination control directly, since a method involving calibration for correctness could reduce hallucination if a suitable domain-specific calibration set were available. The approach of the paper is simple yet innovative. In a basic sense it does the following: In the first step, the method samples from the generative model conditioned on a fixed input. There is a calibration parameter that controls, based on a suitable non-conformity measure, that the generations contain at least one correct generation. Then, the generated set is pruned further using separate calibration parameters based on diversity and factuality considerations. Unlike some previous works, the sequential nature of the process permits the overall admissibility to be easily factorizable, permitting a direct application of conformal prediction proper. An interesting connection is also made to the pareto methods since there are multiple calibration parameters to be handled. The experiments report a general improvement datasets and are sufficient to demonstrate the applicability of the proposed method.

**Strengths:**

-- The paper is quite well-written and clear. Each of the steps are well-described and easy to follow.

-- The method is novel along multiple axes: Wrappers generally make far more simplifying assumptions due to the intractability of conformal prediction directly in such settings; the admissibility control criteria and the connections to pareto methods are interesting and could provide straightforward avenues for future work.

-- While still compute intensive (since multiple generations are required), it could be tuned based on available data for some domain.

**Weaknesses:**

-- There are now a bunch of papers on using conformal wrappers for filtering long form generations by dividing them into segments and then scoring. I think it would have been great to evaluate on such tasks as well, as generally QA type tasks are a bit too easy.

-- Sampling multiple times and expecting a correct response can be quite compute-intensive.

**Questions:**

-- I might have missed it, but do at least a fraction of the experiments report some kind of human evaluation? I think to validate the method, at least some of it might be reasonable. While using another generative model (as a judge or to generate a calibration set) is popular, it still presents an incomplete analysis.

-- The authors might also want to discuss the works on conditional language validity by Cherian https://arxiv.org/abs/2406.09714 and also the earlier work by Mohri and Hashimoto https://arxiv.org/abs/2402.10978. Further, there is also a literature on confidence scoring which is often used for fine-tuning and reducing hallucinations. e.g. Kuhn et al. https://arxiv.org/abs/2302.09664, Lin et al. https://arxiv.org/abs/2305.19187, Wang and Holmes https://web3.arxiv.org/abs/2406.05213. It would be useful to include a brief discussion of these and how conformal methods might be used to calibrate such scores. It would help to bridge the two somewhat separate lines of enquiry together.

---

> ### Author Response · Authors · 2024-11-19
>
> We appreciate the time and effort taken to review our manuscript. We are delighted to hear that you appreciate the novelty and clarity of our work. We address the mentioned weaknesses and questions one-by-one:
>
> > There are now a bunch of papers on using conformal wrappers for filtering long form generations by dividing them into segments and then scoring. I think it would have been great to evaluate on such tasks as well, as generally QA type tasks are a bit too easy.
>
> We thank the reviewer for pointing out these interesting concurrent related works. We are happy to incorporate them into our manuscript and discuss their relationship to ours. We see that these conformal wrappers are beneficial for long text responses. However, a critical assumption for such methods to be practical is that text responses indeed consist of multiple distinct claims with little coherence, so that the returned text response is meaningful and assessable without the removed claims. This assumption may not hold in the context of QA tasks, where responses typically need to be more coherent and holistic and cannot be divided into sub-claims. We incorporated this discussion into our related work section (Section 6).
>
> > Sampling multiple times and expecting a correct response can be quite compute-intensive.
>
> We thank the reviewer for raising this aspect to our attention. This is a good point. We have incorporated an additional paragraph to our discussion section (Appendix I in the revised manuscript), where we discuss this aspect and sketch architecture-specific measures that may reduce the computational cost during inference, in comparison to i.i.d. sampling. During calibration, we note that SCOPE-Gen is vastly more efficient than the most similar existing approach to ours (CLM). We have also added a sentence to elevate this point in our results section (Section 5.2).
>
> We furthermore address the reviewer's questions:
>
> > I might have missed it, but do at least a fraction of the experiments report some kind of human evaluation? I think to validate the method, at least some of it might be reasonable. While using another generative model (as a judge or to generate a calibration set) is popular, it still presents an incomplete analysis.
>
> We agree with the reviewer. However, such an endeavor requires extensive collaboration with domain experts, which is beyond the current scope. The present work is meant as a methodological contribution. We note that the admissibility control guarantee holds irrespective of what (or who) the admissibility function is. We are, however, in the process of initiating a partnership with medical professionals to apply our method in real-world settings involving human evaluations. We plan to explore this in future work and believe that our current study lays the essential groundwork for these applications.
>
> > The authors might also want to discuss the works on conditional language validity by Cherian https://arxiv.org/abs/2406.09714 and also the earlier work by Mohri and Hashimoto https://arxiv.org/abs/2402.10978. Further, there is also a literature on confidence scoring which is often used for fine-tuning and reducing hallucinations. e.g. Kuhn et al. https://arxiv.org/abs/2302.09664, Lin et al. https://arxiv.org/abs/2305.19187, Wang and Holmes https://web3.arxiv.org/abs/2406.05213. It would be useful to include a brief discussion of these and how conformal methods might be used to calibrate such scores. It would help to bridge the two somewhat separate lines of enquiry together.
>
> We thank the reviewer for pointing out these works to us. As mentioned earlier, we have included the mentioned works about conformal wrappers that chunk long answers into smaller claims in the revised version of our manuscript (see Section 6). Regarding the mentioned non-conformal methods, we decided to incorporate them into our discussion section (now Appendix I) as an additional paragraph called *"Reducing Hallucination without Guarantees"*, because the relation is somewhat speculative and more of a venue for future work.
>
> We sincerely thank the reviewer for the insightful comments, which have helped us present a more comprehensive overview of the field.

---

> > ### Comment · Reviewer_XFHD · 2024-11-25
> > **Thanks**
> >
> > Thanks for the comment. If accepted, please do include a wider discussion in the main body of the paper.

---

### Author Response · Authors · 2024-11-19
**General Response**

We thank all reviewers for their time and effort in reviewing our manuscript. The reviewers found our approach to be practically relevant (*"The paper addresses an important problem"* (fuTF)) and efficient (*"This paper presents an efficient approach for generating prediction sets with admissibility guarantees"* (WrdN)) and it was regarded to be well-written (*"Generally well-written"* (7ufb)), with reviewers stating that it is *"novel along multiple axes", "simple yet innovative"* (XFHD) and that claims are experimentally supported (*"Experimental results support the method’s effectiveness"* (WrdN)). We highly value the constructive feedback provided by all reviewers, and incorporated their suggestions in the revised manuscript to further improve our work.

In summary, the main contribution of our work is to introduce Sequential Conformal Prediction for Generative Models (SCOPE-Gen), a sequential conformal prediction method producing prediction sets that satisfy a rigorous statistical guarantee called conformal admissibility control for black-box generative models. In comparison to prior work, our method demonstrates a large reduction in the number of admissibility evaluations during calibration. This reduction is important in safety-critical applications, where these evaluations must be conducted manually by domain experts and are therefore costly and time consuming.

We took all the reviewers’ comments into account and made the following main changes (highlighted in blue in the revised manuscript):

1. We incorporated an additional paragraph called "Computational Demand vs. Generality." into our discussion section (Appendix I in the revised manuscript) that discusses the computational demand of SCOPE-Gen during inference, in comparison to standard i.i.d. sampling. Additionally, we highlighted the vast computational benefit of SCOPE-Gen in comparison to CLM during calibration in Section 5.2.

2. We added a paragraph called "Detecting Hallucination without Guarantees." into our discussion section (Appendix I in the revised manuscript) that discusses work on uncertainty quantification in language models without providing guarantees. We briefly discuss how distribution-free risk control and such heuristic approaches could lead to promising opportunities for cross-pollination in future work.

3. We elevated the reference to ablation studies in the appendix that feature ablations such as using no filters at all, filters in flipped order and different non-conformity update functions (Appendix F in the revised manuscript). We furthermore incorporated an additional ablation experiment that assesses SCOPE-Gen, all of its ablations and CLM and its ablation for three different choices of admissibility levels $\alpha \in \\{ 0.3, 0.35, 0.4 \\}$.

4. We included a section that describes an approach for post-evaluating the admissibility of SCOPE-Gen, including a corresponding experimental demonstration on all of our data sets (Appendix H).

5. We incorporated a rigorous definition of the empirical quantile function used for conformal prediction (Appendix B).

6. We integrated recent advances on conformal wrappers for language models into our related work (Section 6), with a brief discussion.

Once again, we sincerely thank all reviewers for their thoughtful and constructive comments, which have helped us improve the quality of our work.

---

### Meta-Review · Area_Chair_UKAZ · 2024-12-20

**Metareview:**

This paper propose Sequential Conformal Prediction for Generative Models (SCOPE-Gen), that utilise conformal prediction techniques that operates sequentially, to obtain guarantees and controls for generative models. The proposed approach not only provide practical algorithmic guidelines, but also include interesting Markovian interpretations, well supported by improved experimental results. The reviewers reach the consensus that the contribution is significant and the paper is well-written. It is suggested to include the discussion in the revised manuscript and recommended to appear in ICLR conference.

**Additional Comments On Reviewer Discussion:**

Most of the reviewers' concerns are addressed, and additional discussions are updated to the manuscript accordingly.

---

### Decision · Program_Chairs · 2025-01-22

Accept (Poster)